# Crosslingual Transfer Learning for Low-Resource Languages Based on Multilingual Colexification Graphs

**Yihong Liu**[*◇], **Haotian Ye**[*◇], **Leonie Weissweiler**[*◇], **Renhao Pei**[*], and **Hinrich Schütze**[*◇]

[*]Center for Information and Language Processing, LMU Munich
[◇]Munich Center for Machine Learning (MCML)
`{yihong, yehao, weissweiler}@cis.lmu.de`

## Abstract

In comparative linguistics, colexification refers to the phenomenon of a lexical form conveying two or more distinct meanings. Existing work on colexification patterns relies on annotated word lists, limiting scalability and usefulness in NLP. In contrast, we identify colexification patterns of more than 2,000 concepts across 1,335 languages directly from an unannotated parallel corpus. We then propose simple and effective methods to build multilingual graphs from the colexification patterns: **ColexNet** and **ColexNet+**. ColexNet's nodes are concepts and its edges are colexifications. In ColexNet+, concept nodes are additionally linked through intermediate nodes, each representing an ngram in one of 1,334 languages. We use ColexNet+ to train $\overrightarrow{\text{ColexNet+}}$, high-quality multilingual embeddings that are well-suited for transfer learning. In our experiments, we first show that ColexNet achieves high recall on CLICS, a dataset of crosslingual colexifications. We then evaluate $\overrightarrow{\text{ColexNet+}}$ on roundtrip translation, sentence retrieval and sentence classification and show that our embeddings surpass several transfer learning baselines. This demonstrates the benefits of using colexification as a source of information in multilingual NLP.

## 1 Introduction

Multilingual representations are beneficial in natural language processing (NLP) due to their ability to transfer knowledge across languages (Artetxe and Schwenk, 2019; Conneau et al., 2020; Fan et al., 2021). Typically, such representations are learned through pre-training Large Language Models (LLMs) (Brown et al., 2020; Chowdhery et al., 2022; Touvron et al., 2023) or multilingual word embeddings (Ammar et al., 2016; Lample et al., 2018; Dufter et al., 2018). However, LLMs require enormous amounts of data to train, limiting their use mostly to high-resource and medium-resource

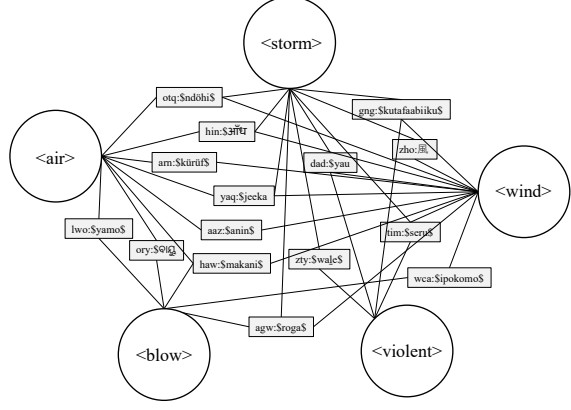

Figure 1: A subgraph of ColexNet+. Circles (<air>, <storm>, . . . ): concept nodes. Rectangles (otq:$ndöhi$, lwo:$yamo$, . . . ): ngram nodes (each prefixed by its ISO 693-3 code). Ngram nodes realize colexifications, e.g., <air> and <storm> are linked, through translations in the parallel corpus, to the Querétaro Otomí ngram "ndöhi" (i.e., otq:$ndöhi$).

languages (Zhou et al., 2023). Alternatively, multilingual word embeddings are widely used in NLP because of their simplicity and good performance (Ammar et al., 2016; Lample et al., 2018; Jawanpuria et al., 2019). However, most existing multilingual embeddings are learned through *word-context* information, without leveraging global cooccurrence information in individual languages or across languages, which can help distinguish distinct meanings conveyed by a lexical form. Therefore, we see a pressing need in NLP for massively multilingual word embeddings that span a large number of languages (1,335 in our case) and that specifically account for global occurrence and are a good basis for crosslingual transfer learning.

*Colexification* has gained increasing attention in comparative linguistics and crosslingual NLP. According to François (2008), a language *colexifies* two distinct meanings if it expresses them with the same lexical form. Different languages have different *colexification patterns*. For example, while English has separate words for <hand>

and <arm>[1], Russian 'рука' colexifies these two concepts. Most prior work explores colexification (Floyd et al., 2021; Brochhagen and Boleda, 2022; List, 2023) using manually curated crosslingual datasets that consist of multilingual word lists such as CLICS (List, 2018; List et al., 2018; Rzymski et al., 2020). However, relying on these datasets has several limitations: extension to more languages and more concepts can be challenging; these datasets contain lists of lemmata and (in a corpus-based approach for low-resource languages without morphological resources) cannot easily be used for the processing of occurrences in context.

To overcome these limitations and boost crosslingual transfer learning especially for low-resource languages, we use the **P**arallel **B**ible **C**orpus (PBC) (Mayer and Cysouw, 2014), which has verse-level aligned translations of the Bible in 1,335 languages, to identify colexification patterns (a verse in PBC roughly corresponds to a sentence). With the identified patterns between a wide range of concepts, we propose novel algorithms that efficiently build large-scale multilingual graphs. To the best of our knowledge, this is the first work that constructs graphs of colexification and trains multilingual representations for crosslingual transfer learning directly from a parallel corpus on a large scale. We show that the graphs capture the links between concepts across languages and that the derived multilingual representations considerably improve crosslingual transfer on downstream tasks. Previous work on building monolingual graphs (Jauhar et al., 2015; Ustalov et al., 2017) or multilingual graphs (Harvill et al., 2022; Jafarinejad, 2023; Chen et al., 2023) is different as it (1) does not consider words in context and only uses lemmata, (2) is based on external sense inventories such as WordNet (Miller, 1995) and BabelNet (Navigli and Ponzetto, 2012; Navigli et al., 2021), which are not available for many low-resource languages, and (3) does not investigate the crosslingual transferability of the multilingual representations on NLP downstream tasks such as sentence retrieval or classification in a crosslingual scenario.

The contributions of this work are as follows: (i) We present ColexNet, a graph of concepts based on colexification patterns that are directly extracted from a parallel corpus. (ii) By extending ColexNet, we further present ColexNet+, a large-scale multi-lingual graph that additionally contains ngrams in 1,334 languages that instantiate those patterns. (iii) We contribute to crosslingual transfer learning, by using ColexNet+ to generate multilingual embeddings: $\overrightarrow{\text{ColexNet+}}$. We show that $\overrightarrow{\text{ColexNet+}}$ outperforms several baselines on roundtrip translation, verse retrieval, and classification. (iv) We evaluate ColexNet on CLICS and show that we identify a large portion of the ground-truth colexifications. (v) Going beyond many works on crosslingual transfer that focus on transfer from English, we systematically investigate the effect of the source language on successful transfer with $\overrightarrow{\text{ColexNet+}}$: we use 1,245 languages as sources and experiment on $1{,}245 \times 1{,}245$ transfer directions. (vi) We make our code, graphs, and embeddings publicly available.[2]

## 2 Related Work

There are many ways to learn multilingual word embeddings. One common way is to first learn monolingual embeddings on each language separately through, e.g., Word2Vec (Mikolov et al., 2013), GloVe (Pennington et al., 2014), or fastText (Bojanowski et al., 2017), and then map them into the same space (Artetxe et al., 2017; Lample et al., 2018; Artetxe et al., 2018). Another group of methods uses parallel corpora to directly learn bilingual embeddings (Hermann and Blunsom, 2014; Chandar et al., 2014; Levy et al., 2017). Our work is related to that of Dufter et al. (2018), which also learns embeddings on the PBC, but does not take advantage of colexification, i.e., the explicit modeling of relations between colexified concepts/ngrams. We use S-ID (Levy et al., 2017) and embeddings from Dufter et al. (2018) as baselines.

One of the best-known and widely used multilingual resources is BabelNet (Navigli and Ponzetto, 2012; Navigli et al., 2021). BabelNet has been used for learning or enhancing embeddings (Iacobacci et al., 2015; Camacho-Collados and Pilehvar, 2018; Conia and Navigli, 2020; Levine et al., 2020; Harvill et al., 2022; Chen et al., 2023) for lexical-level tasks such as semantic word similarity and word sense disambiguation (Speer and Lowry-Duda, 2017; Conia and Navigli, 2020; Procopio et al., 2021; Navigli et al., 2022). Our focus is on the coverage of many more languages (i.e., larger scale in terms of languages) for crosslingual transfer learning. While hand-curated lexica often have

---

[1] We represent a concept (from any language) by surrounding an English word that refers to the concept with "<>".

[2] https://github.com/cisnlp/ColexificationNet

better quality than automatically learned resources, they are not available for most of our languages. Ultimately, the two approaches should be combined.

Colexification was introduced by Haspelmath (2003) in the context of grammatical semantics. François (2008) then used colexification as the foundation for studying semantic change crosslinguistically. CLICS (List, 2018; List et al., 2018; Rzymski et al., 2020) is a crosslingual database that facilitates research on colexification. Languages can differ in their colexification patterns, which are influenced by many factors such as human cognition, language family, and geographic area (Jackson et al., 2019; Xu et al., 2020; Segerer and Vanhove, 2022). An empirical study by Bao et al. (2021) indicates that no pair of concepts is colexified in every language. On the other hand, a recent investigation on conceptualization from the PBC shows some concepts are more likely to be involved in colexification than others (Liu et al., 2023). Such universal colexification patterns across languages reflect crosslinguistic similarities (Youn et al., 2016; Georgakopoulos et al., 2022). Therefore, by integrating colexification patterns of as many languages as possible, we can generate multilingual representations that are suitable for massively crosslingual transfer learning.

## 3 Methodology

### 3.1 Data

We use 1,335 Bible translations from the PBC corpus (Mayer and Cysouw, 2014). Each translation is from a different language (identified by its ISO 639-3 code). Prior work (Asgari and Schütze, 2017; Dufter et al., 2018; Weissweiler et al., 2022) has used subsets of the corpus. In contrast, we follow Conceptualizer (Liu et al., 2023) and use all parallel verses between English and other languages. This gives us better coverage of concepts and the contexts in which they occur.

### 3.2 Colexification pattern identification

**Concept Pool.** Conceptualizer (Liu et al., 2023) uses a small manually selected group of **focal concepts**, i.e., concepts of interest (83 in total) and constructs a set of strings to represent each concept. For example, it uses {`$belly$`, `$bellies$`} to represent the focal concept <belly>, where `$` is the word boundary. Manually defining the sets is not feasible when a large number of concepts are to be explored. Thus, in this work, we lemmatize the English corpus and regard each lemma as a concept. The set of all lemmata forms the concept pool $F$.

**Conceptualizer.** Conceptualizer (Liu et al., 2023) creates a bipartite directed alignment graph between source language concepts and target language strings. It consists of a forward pass (FP) and a backward pass (BP). This kind of two-step workflow is also used in extracting semantic relations (Dyvik, 2004) and paraphrases (Bannard and Callison-Burch, 2005) from bilingual parallel corpora. A key difference compared with this prior work is that Conceptualizer works on the ngram level instead of the token level; this facilitates the extraction of any associations hidden inside words. In Conceptualizer, FP first searches for target language ngrams highly associated with a given focal concept; BP then searches for English ngrams highly correlated with the target ngrams identified in FP. The association is measured using $\chi^2$ score. The process can detect if the conceptualization of the focal concept diverges in any language. For example, starting from concept <hand>, FP finds the Russian ngram 'рук', and BP then finds two English ngrams 'hand' and 'arm'. This indicates that the conceptualization of these concepts diverges in English and Russian. The divergence of conceptualization in the lexical forms indicates a difference in their colexification patterns: Russian colexifies the concepts <hand> and <arm> (in the word 'рук') while English does not.

**Forward Pass.** Let $f$ be a focal concept in $F$ and $V_f$ the set of verses in which $f$ occurs. FP identifies a set of ngrams $T$ in target-language $l$ where each ngram can refer to concept $f$, i.e., $T = \text{FP}(f, l)$. We exhaustively search all ngrams $t$ within all tokens[3] in the parallel corpus in target language $l$ for high correlation with $V_f$. This procedure is similar to Östling and Kurfalı (2023)'s subword-level alignment, but we align concepts in English and subwords in other target languages. E.g., we start from <hand> and find that the Russian ngram 'рук' has the highest correlation with $V_{<hand>}$, which indicates 'рук' can refer to <hand>. Like Conceptualizer, we use $\chi^2$ as a measure of correlation and iterate FP until the cumulative coverage $T = \bigcup t$ of focal concept $f$ exceeds a threshold $\alpha = 0.9$, but for a maximum of $M = 3$ iterations. See §A for a discussion of these hyperparameters.

---

[3]Similar to the setting in Conceptualizer (Liu et al., 2023), we use `$` to denote token boundaries; e.g., `$k`, `$ke`, `$ke$`, `k`, `ke`, `ke$`, `e`, `e$` are all valid ngrams of token `$ke$`.

**Backward Pass.** BP is essentially the same as FP, but the search direction is reversed. Let $V_T$ be the set of verses in which at least one ngram in $T$ (identified in FP for target language $l$ and concept $f$) from target language $l$ occurs. We exhaustively search all concepts $c$ from the concept pool $F$ for high correlations with $V_T$. Let $C = \mathrm{BP}(T, l)$ be the final set of identified concepts. If $|C| = 1$ and $c \in C \wedge c = f$, this indicates the ngrams can only refer to the concept $f$ according to the bilingual context. Alternatively, if $|C| > 1$, this indicates language $l$ colexifies concepts in $C$ through ngrams $T$. For example, by performing BP on ngram 'рук', we get <hand> and <arm> as the result, which indicates Russian colexifies <hand> and <arm>. Notably, since we consider ngrams instead of tokens on the target language side, this allows us to also identify *partial colexification* patterns in BP, i.e., patterns that do not involve an entire word, but rather part of it. We show such examples in §B.

### 3.3 ColexNet

We run FP and BP for all 1,806 focal concepts in the English concept pool $F$ that have frequency between 5 and 2000 and for every language $l$ in our set of 1,334 target languages $L$ (excluding English). This allows us to uncover the colexification patterns in 1,334 languages. We formalize the relations of the colexification patterns as an undirected graph, where each node is a concept represented by an English lemma and each edge indicates that at least one language colexifies the two connected concepts. Formally, let $\mathcal{G}(F, \mathcal{E}, w_c, w_n)$ be a weighted undirected graph on vertices $F$, i.e., the concept pool, where $\mathcal{E}$ is a set of undirected edges; $w_c$ is an edge weighting (counting) function: $F \times F \to \mathbb{Z}_+$, which returns, for a pair of concepts, the number of languages colexifying them; $w_n$ is an edge record function, which returns all ngrams that colexify a given pair of concepts. We show the graph construction in Algorithm 1.

In this study, we use a threshold $\lambda$ to control the confidence of the colexification edges: we remove an edge $e$ if $w_c(e) < \lambda$. The intuition is that: if two concepts $f_1$ and $f_2$ are colexified in many languages, we can be more certain that the edge between $f_1$ and $f_2$ is correctly identified. Looking at it the other way around, if two concepts are only colexified in a few languages, this might be a wrongly identified pattern because of verse-level misalignment, free translation, or other errors in

---

**Algorithm 1:** ColexNet & ColexNet+

**Input:** set of languages $L$, concept pool $F$, minimum number of languages threshold $\lambda$;
**Output:** ColexNet $\mathcal{G}_1$, ColexNet+ $\mathcal{G}_2$;

1  $\mathcal{G}_1 \leftarrow$ graph with $F$ as nodes and no edges $\mathcal{E}_1$;
2  $\mathcal{G}_2 \leftarrow$ graph with no nodes $V$ and no edges $\mathcal{E}_2$;
3  $V \leftarrow F$;
4  set $w_c(\cdot) = 0$ and $w_n(\cdot) = \emptyset$ by default;
5  **for** $l \in L$ **do**
6      **for** $f \in F$ **do**
7          $T \leftarrow \mathrm{FP}(f, l)$;
8          $C \leftarrow \mathrm{BP}(T, l)$;
9          $V \leftarrow V \cup T$;
10         **for** $c \in C$ **do**
11             $\mathcal{E}_1 \leftarrow \mathcal{E}_1 \cup (f, c)$;
12             $w_c((f, c)) += 1$;
13             $w_n((f, c)) \leftarrow w_n((f, c)) \cup T$;
14         **end**
15     **end**
16 **end**
17 **for** $e \in \mathcal{E}_1$ **do**
18     **if** $w_c(e) < \lambda$ **then**
19         remove $e$ from $\mathcal{E}_1$;
20     **end**
21 **end**
22 **for** $e \in \mathcal{E}_1$ **do**
23     $(f_1, f_2) \leftarrow e$;
24     **for** $v \in w_n((f_1, f_2))$ **do**
25         $\mathcal{E}_2 \leftarrow \mathcal{E}_2 \cup (f_1, v) \cup (v, f_2)$;
26     **end**
27 **end**
28 remove nodes in $\mathcal{G}_1$ and $\mathcal{G}_2$ that have zero degree;
29 **return** $\mathcal{G}_1$, $\mathcal{G}_2$

---

the data. See Table 4 for the influence of different $\lambda$ on graph statistics. In addition, we remove zero-degree nodes to filter out isolated concepts.

### 3.4 ColexNet+

ColexNet only contains concepts that are expressed in English lemmata and cannot be directly used to learn multilingual representations for the target languages. Therefore, we propose ColexNet+, a large multilingual graph expanded from ColexNet by including target language ngrams that instantiate the colexification patterns identified in ColexNet. Specifically, we replace each edge $(f_1, f_2)$ with a set of new pairs of edges: (1) find the set of ngrams $w_n((f_1, f_2))$ that colexify concepts $f_1$ and $f_2$ (in any language) and (2) for each ngram $v$ in the set, insert new edges $(f_1, v)$ and $(v, f_2)$. To define a clean bipartite structure, we do not include the original $(f_1, f_2)$; this guarantees that only concept-ngram edges and no concept-concept edges occur. In addition, any two related concepts (i.e., there is an edge connecting the two concepts in ColexNet) are always implicitly connected through ngram nodes in ColexNet+ that associate the two con-

cepts. Figure 1 shows a subnetwork of ColexNet+ consisting of a few concepts and ngrams in different languages that colexify them. The graph construction is shown in Algorithm 1.

As ColexNet+ is expanded from ColexNet, this allows us to only include pairs of edges expanded from reliable edges ($w_c(e) \geq \lambda$) in ColexNet. The number of nodes and edges included in ColexNet+ is thus influenced by $\lambda$. The higher $\lambda$, the fewer nodes and edges will be in ColexNet+. §A presents statistics and performance for different values of $\lambda$.

### 3.5 Multilingual Embedding learning

To capture the semantic relations among the nodes and the structure of ColexNet+, we use Node2Vec (Grover and Leskovec, 2016) to generate node representations. Let $v$ be the node that a random walk currently resides in, $t$ the node that the walk has traversed in the last step, and $x$ the node that the walk will visit in the next step. Node2Vec calculates the unnormalized transition probability from $v$ to $x$ as $\pi_{vx} = \alpha_{pq}(t, x) \cdot w((v, x))$ for sampling the next node $x$ in the graph, where $w((v, x))$ is the weight of the undirected edge $(v, x)$,

$$\alpha_{pq}(t, x) = \begin{cases} \frac{1}{p} & \text{if } d_{tx} = 0 \\ 1 & \text{if } d_{tx} = 1 \\ \frac{1}{q} & \text{if } d_{tx} = 2 \end{cases}$$

and $d_{tx}$ is the shortest path distance between $t$ and $x$. The transition probability determines if either a new node or an already-visited node (regardless of a concept or ngram node) will be sampled. In ColexNet+, $d_{tx} \neq 1$ for any nodes $t$ and $x$, because a concept (resp. ngram) node will not connect with other concept (resp. ngram) nodes. We set return parameter $p = .5$ and in-out parameter $q = 2$ in the hope of encoding more "local" information, as this setting approximates breadth-first sampling according to Grover and Leskovec (2016).

Below, we show that the multilingual representations trained this way have some desirable properties, e.g., representations of ngrams from different languages that refer to the same concept can be highly cosine-similar, which is important for zero-shot crosslingual transfer learning.

## 4 Experiments

To evaluate our proposed methods, we conduct the following experiments: (1) colexification identification; (2) roundtrip translation; (3) verse retrieval; and (4) verse classification. Experiment (1)

evaluates the colexification patterns we identify in ColexNet. Experiments (2), (3) and (4) evaluate the learned multilingual embeddings $\overrightarrow{\text{ColexNet+}}$.

### 4.1 Baselines

To evaluate the effectiveness of $\overrightarrow{\text{ColexNet+}}$, our multilingual embeddings, we consider several previously proposed strong multilingual embeddings as baselines for downstream tasks. The dimension of all embeddings (ours and the baselines) is set to 200 for a fair comparison. In addition, we consider three non-embedding baselines: bag-of-words (BOW), XLM-R (Conneau et al., 2020) and Glot500-m (ImaniGooghari et al., 2023). The first is a random baseline and is expected to perform the worst because a BOW model is only trained on the English corpus, which does not directly transfer to other languages. The latter two are strong multilingual pretrained models. XLM-R is pretrained on 100 languages while Glot500-m is a continued-pretrained version of XLM-R on the Glot500-c corpus (ImaniGooghari et al., 2023) that includes more than 500 languages. We choose the base version of these multilingual pretrained models. We introduce the embedding baselines below.

**S-ID embedding.** Levy et al. (2017) show that S-ID embeddings, which leverage the sentence ID feature, are effective in learning good multilingual embeddings from parallel corpora. We use pairs of a verse ID identifier and a token in this verse as input to Word2Vec (Mikolov et al., 2013) to train S-ID embeddings. For example,, the pairs (01049027, Wolf) and (01049027, 狼) will be presented in the data because '狼' (resp. 'Wolf') occurs in Chinese (resp. German) in verse number 01049027. This is a strong baseline because the verse number is an abstract representation of the context. Therefore it encourages words occurring across languages in the same verse to have similar embeddings.

**CLIQUE & $N(t)$ embedding.** CLIQUE embeddings (Dufter et al., 2018) are learned on cliques extracted from PBC. Each clique is a set of tokens from different languages that refer to the same concept. The embeddings are then learned from token-clique pairs. Additionally, to take the connections between concepts into account, Dufter et al. (2018) consider the neighbors (tokens that are connected with the current node in the dictionary graph) of each token and train embeddings on those pairs of neighbors, which we refer to as $N(t)$ embedding.

**Eflomal-aligned embedding.** We construct an alignment graph of words by using Eflomal (Östling and Tiedemann, 2016) and learn embeddings on the graph as another strong baseline. Specifically, we align the English Bible with Bibles in all other target languages. We define the edge set of the graph as the set of all edges that connect an English word with its aligned target language word (if there are at least two such alignments). Finally, we use Node2Vec (same hyperparameters as for ColexNet+) to learn multilingual embeddings.

## 4.2 Colexification identification

We first evaluate how well ColexNet performs at identifying colexification patterns. We use CLICS (List, 2018; List et al., 2018; Rzymski et al., 2020), a database of colexifications, as the gold standard. Each node in CLICS is a concept expressed in English. In ColexNet, we use English lemmata as expressions of concepts whereas CLICS also includes short noun phrases. We only consider the common concepts, i.e., concepts that are expressed as English words and occur in both CLICS and ColexNet. For each start concept $s$ in the common concepts $P$, let $T(s)$ be the neighbors in CLICS, i.e., a set of concepts that have a colexification relation with $s$ and $C(s)$ be the neighbors in ColexNet. Then we compute the recall for $s$ as $|T(s) \cap C(s)|/|T(s)|$. To have a global view of the performance, we report the micro average recall (MicroRec.):

$$\text{MicroRec.} = \frac{\sum_{s \in P} |T(s) \cap C(s)|}{\sum_{s \in P} |T(s)|}$$

macro average recall (MacroRec.):

$$\text{MacroRec.} = \sum_{s \in P} \frac{|T(s) \cap C(s)|}{|T(s)||P|}$$

and the average number of incorrect (not present in CLICS) colexifications per concept (#aw_colex):

$$\text{\#aw\_colex} = \sum_{s \in P} \frac{|C(s) - T(s)|}{|P|}$$

#aw_colex has a similar function to precision. We do not use precision as a measure because it can underestimate the performance, as many patterns included in ColexNet can actually be correct (see §B for examples). The #aw_colex measure can better and more directly reflect how the exact number of "incorrect" patterns per concept changes with respect to the value of $\lambda$. Results are shown in Table 1.

| $\lambda$ | $P$ | MicroRec. | MacroRec. | #aw_colex |
|---|---|---|---|---|
| 1 | 1220 | 0.71 | 0.80 | 580.87 |
| 5 | 1056 | 0.63 | 0.77 | 84.34 |
| 10 | 1001 | 0.58 | 0.73 | 42.04 |
| 20 | 935 | 0.54 | 0.70 | 22.91 |
| 50 | 833 | 0.48 | 0.66 | 10.69 |
| 100 | 761 | 0.42 | 0.62 | 5.78 |

Table 1: The number of common concepts ($P$) in ColexNet and CLICS, Micro and Macro average recall, as well as the average number of colexification patterns per concept that are not in CLICS (#aw_colex) for different thresholds $\lambda$ (the minimum number of languages for keeping a colexification edge).

If the constraint $\lambda$ is stricter, fewer concepts and fewer edges (both colexification edges contained and not contained in CLICS) will be included in ColexNet. Thus, we observe a consistent drop in both micro and macro recall. On the other hand, we observe a decrease in #aw_colex when we increase $\lambda$, as CLICS edges are less likely to be removed than edges missing from CLICS: many languages can share the same colexification patterns for some concepts whereas edges not in CLICS will not be shared across many languages. This can be verified by the steepness of the decrease in #aw_colex. From $\lambda = 1$ to 5, around 500 edges not in CLICS are removed for each concept. When $\lambda > 5$, the speed decreases, suggesting the identified colexification edges are more reliable. In summary, high recall indicates that we successfully identify many ground-truth colexifications directly from PBC. It is important to note that CLICS' coverage is far from complete for low-resource languages: for many of them, fewer than 100 concepts are included in CLICS. Therefore, #aw_colex gives some indication of performance or discrepancy between CLICS and ColexNet, but many of the edges not in CLICS are actually correct. On the other hand, ColexNet is not immune to semantic errors (Peirsman and Padó, 2008), such as antonyms, hypernyms, or hyponyms, due to co-occurrence or free translation. See §B for a detailed analysis of the identified colexifications.

## 4.3 Roundtrip translation

We additionally use roundtrip translation (Dufter et al., 2018) to assess the quality of multilingual representations. Let $[l_0, l_1, l_2, ..., l_R]$ be a sequence of languages where $l_0 = l_R$ is the source language and $l_i \neq l_0 \forall 1 \leq i \leq R-1$ different intermediate languages. Roundtrip translation starts with a word $w_0$ in $l_0$ and iteratively finds the word $w_r$ in

| | roundtrip translation | | | verse retrieval | | | verse classification |
|---|---|---|---|---|---|---|---|
| | top-1 | top-5 | top-10 | top-1 | top-5 | top-10 | |
| BOW | - | - | - | 0.02 | 0.05 | 0.06 | 0.09 |
| S-ID (Levy et al., 2017) | 0.10 | 0.21 | 0.25 | 0.17 | 0.29 | 0.35 | 0.32 |
| CLIQUE (Dufter et al., 2018) | 0.22 | 0.63 | 0.79 | 0.41 | 0.62 | 0.70 | 0.44 |
| $N(t)$ (Dufter et al., 2018) | 0.22 | 0.53 | 0.65 | 0.28 | 0.46 | 0.55 | 0.47 |
| XLM-R (Conneau et al., 2020) | - | - | - | 0.04 | 0.07 | 0.09 | 0.15 |
| Glot500-m (ImaniGooghari et al., 2023) | - | - | - | 0.11 | 0.17 | 0.21 | 0.22 |
| Eflomal-aligned | 0.24 | 0.58 | 0.70 | 0.61 | 0.76 | 0.81 | 0.48 |
| $\overrightarrow{\text{ColexNet+}}$ | **0.44** | **0.85** | **0.93** | **0.65** | **0.80** | **0.84** | **0.49** |

Table 2: Results of different multilingual embeddings on roundtrip translation, verse retrieval, and verse classification tasks. Each number for roundtrip translation (top-$k$ accuracy, $k \in \{1, 5, 10\}$) is the average of 10 runs with 3 randomly selected intermediate languages. Each number in verse retrieval (top-$k$ accuracy, $k \in \{1, 5, 10\}$) and verse classification (macro $F_1$) is the average over all available languages (1,250 for verse retrieval, 1,245 for verse classification). We also report the performance of BOW, XLM-R, and Glot500-m. The first serves as a random baseline whereas the latter two are strong multilingual model baselines. The performance reported for Glot500-m is different from the original paper as it was evaluated on a subset of languages the model supports, while we evaluate the model on all languages that $\overrightarrow{\text{ColexNet+}}$ supports, making comparison easier. **Bold**: best result per column.

language $l_r$ ($1 \leq r \leq R$) that is closest to word $w_{r-1}$ in language $l_{r-1}$ in the embedding space. If $w_0 = w_R$, this indicates that the $R - 1$ "intermediate" words have representations similar to $w_0$ and represent the meaning of $w_0$. We compute the percentage of roundtrips for $w_0$ that are successful, i.e., $w_0 = w_R$ (top-1 accuracy). In addition, we also report top-5 and top-10 accuracies (i.e., $w_0$ is in the $k$ ($k = 5$ or $k = 10$) nearest neighbors). We set $R = 4$, $l_0 = $ English and take 1,654 English words that occur in all embedding spaces as the starting point $w_0$. For each trial, we randomly select three intermediate languages and then compute results for each of the 1,654 $w_0$. We run this experiment ten times and report averages. We ensure that the intermediate languages are different in each trial.

### 4.4 Verse retrieval

Similarly to Glot500-m, we use 500 English-aligned verses from PBC for verse retrieval. 1,250 languages are used (we remove 85 languages that cover fewer than 400 out of the 500 verses). We represent each verse as the average of the embeddings of its units. Given a verse $v_e$ in English, we find the most cosine-similar verses $v_l$ in target language $l$. We then compute top-1, top-5 and top-10 accuracy for the returned ranking (i.e., the correct verse is in the top-1, top-5, top-10 nearest neighbors) and average first over verses and then languages.

### 4.5 Verse classification

We evaluate our multilingual embeddings on Taxi1500 (Ma et al., 2023). It provides 860/106/111

verses for train/valid/test sets in more than 1,500 languages. Each verse is annotated with one of six classes: 'recommendation', 'faith', 'description', 'sin', 'grace', and 'violence'. We use a subset of 1,245 languages, those covered by both Taxi1500 and $\overrightarrow{\text{ColexNet+}}$. We perform zero-shot transfer by training a logistic classifier on English train and evaluating on the test set of the other 1,244 languages. Similar to verse retrieval, we represent a verse as the average of its embeddings. We report macro $F_1$, first averaged over verses (per language) and then averaged over languages.

### 4.6 Results & Discussion

Table 2 shows that $\overrightarrow{\text{ColexNet+}}$ BOW, S-ID, CLIQUE, $N(t)$, Glot500-m and Eflomal-aligned on all three tasks: roundtrip translation, verse retrieval, and verse classification. $\overrightarrow{\text{ColexNet+}}$ shows a large improvement over the baselines, especially for roundtrip translation and verse retrieval. The bad performance of BOW is expected, as previously mentioned, because the English vocabulary does not necessarily transfer to other languages. $\overrightarrow{\text{ColexNet+}}$'s improvement over S-ID is probably due to the fact that there is only verse ID information provided to serve as verse-level context information in S-ID. Token-level alignment information, however, is not available to S-ID. In other words, using abstract context identifiers alone cannot provide enough information to learn good multilingual embeddings for crosslingual transfer. When comparing $\overrightarrow{\text{ColexNet+}}$ with XLM-R and Glot500-m, we see a clear improvement in either verse retrieval

or verse classification. The major reason is that both XLM-R and Glot500-m are not trained on all languages that are supported by $\overrightarrow{\text{ColexNet+}}$. Due to a lack of data in some low-resource languages, it is difficult to train a good language model in those languages. In contrast, $\overrightarrow{\text{ColexNet+}}$ demonstrates the possibility of multilingual embeddings: with a small multilingual corpus where we can extract colexifications, it is already enough to support large-scale zero-shot transfer for the low-resource languages by training embeddings.

CLIQUE, $N(t)$, and Eflomal-aligned achieve similar performance on roundtrip translation (top-1). However, when $k$ becomes larger ($k = 5$ or 10), we see that CLIQUE performs better than $N(t)$ and Eflomal-aligned. This is not surprising, since CLIQUE specifically creates cliques of tokens that are translations of each other in different languages. Therefore the representations of translations should be similar. Eflomal-aligned also achieves good performance on roundtrip translation when $k$ is large ($k = 5$ or 10) and very close performance to $\overrightarrow{\text{ColexNet+}}$ in verse retrieval / classification. There are a few possible explanations. First, the word alignments are noisy in Eflomal-aligned because it operates on the token level and any information hidden inside each token (i.e., ngrams inside each token) cannot be extracted and utilized (see the discussion also in Liu et al. (2023)). Therefore, by increasing $k$ in roundtrip translation, the influence of such alignment noise is offset, resulting in better results. Second, as we use the average of embeddings of tokens in a verse as the verse representation in verse retrieval / classification; this can mitigate the impact of unimportant tokens.

For verse classification, we find that different embeddings achieve similar performance except for S-ID. On the one hand, this phenomenon indicates that S-ID, though it learns from abstract context information, cannot align words from different languages that refer to the same concepts well, thus preventing transfer from English to low-resource languages. On the other hand, it might indicate that classification is a less difficult task: it does not require the model to have equally good alignment for all concepts as the model can achieve good results just by aligning important concepts. Nevertheless, $\overrightarrow{\text{ColexNet+}}$ still achieves better results than other baselines, suggesting it has better zero-shot transferability. See §E for complete results.

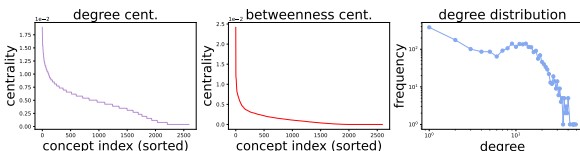

Figure 2: Degree centrality (left), betweenness centrality (middle) and degree distribution (right) of ColexNet.

## 5 Analysis

### 5.1 Analyses on ColexNet

**Basic statistics.** We find that ColexNet has one very large connected component along with a few small connected components. See a visualization of the largest community in ColexNet ($\lambda = 50$) in Figure 4 in the Appendix. Therefore, in the largest community, there is always a path in the colexification graph between two concepts even if they are less related. Figure 2 shows degree/ betweenness centrality and degree distribution of ColexNet. From the figure, we can infer that the connectivity can be attributed to (1) a small group of concepts that are involved in many colexification patterns and (2) a small group of edges serving as "bridges" to connect concepts that are rarely colexified in some languages. Therefore, ColexNet, a graph built by the identified colexification patterns across many languages, approximately forms a small-world or scale-free (Barabási and Bonabeau, 2003) network. See §A.2 for graph-related statistics of ColexNet under different $\lambda$.

**Communities.** We use the Louvain algorithm (Blondel et al., 2008) to find communities in ColexNet. We identify 288 communities. Each community forms a cluster of semantically related concepts. Figure 3 gives the example of community#29: it contains several concepts related to <wind>, <storm> and <wave>. We see that <wind> is often colexified with <blow> (wind blows), with <wave> (waves are caused by wind) and with <violent> (winds can be fierce). At the center of a community, we often find a densely connected clique, indicating their connections are strong in many languages. Some concepts, located at the fringe of the community and connected with one of the densely-connected concepts in the center, are less related to the semantic field of the community and serve as "bridges" to connect with other communities. See §C for further details of the identified communities.

### 5.2 Transfer learning beyond English

NLP research in general and even typological studies are frequently conducted from an English-

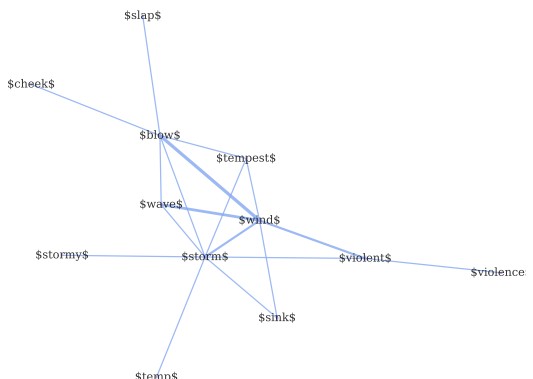

Figure 3: Community #29. Line thickness indicates the number of languages that instantiate a colexification.

|     | verse retrieval | | | verse classification |
| --- | --- | --- | --- | --- |
|     | top-1 | top-5 | top-10 | |
| arb | 0.56 | 0.72 | 0.78 | 0.47 (0.07) |
| rus | 0.55 | 0.72 | 0.78 | 0.48 (0.07) |
| zho | 0.60 | 0.77 | 0.82 | 0.49 (0.05) |
| apn | 0.21 | 0.38 | 0.46 | 0.38 (0.07) |
| muh | 0.22 | 0.39 | 0.48 | 0.38 (0.07) |
| sll | 0.24 | 0.42 | 0.51 | 0.39 (0.08) |
| avg. | 0.46 | 0.64 | 0.71 | 0.44 (0.08) |

Table 3: Verse retrieval/classification for three high-resource languages, the three worst performing languages, and average over all languages (avg.). We also report BOW results for verse classification (in parentheses), which serves as the random baseline. In contrast to the good performance for Arabic (arb), Russian (rus) and Chinese (zho), Apinayé (apn), Mündü (muh) and Salt-Yui (sll) each pose specific difficulties for inducing reliable colexification patterns.

centric perspective. To reduce this bias and further verify our multilingual embeddings' transfer capability, we additionally use *all* available languages (1,245 languages) as the query / train languages for retrieval and classification tasks. To this end, we conduct large-scale experiments that contain $1,245 \times 1,245$ transfer directions. The setup is the same as in §4, where each language takes the role of English as the query / train language. We again represent each verse as the average of the embeddings of its units. For each language, we calculate the average top-$k$ ($k = 1, 5,$ or 10) accuracy for verse retrieval and macro $F_1$ for verse classification over all other languages except the language itself.

In Table 3, we list the transfer performance of three major languages that are typologically different from English: Arabic (arb), Russian (rus), and Chinese (zho); and three languages that achieve **the worst** overall performance: Apinayé (apn), Mündü (muh), and Salt-Yui (sll). See §F for complete results for all languages. For high-resource languages, the performance is close to that achieved for English (see Table 2), indicating that the ngrams are well-aligned and $\overrightarrow{\text{ColexNet+}}$ has good transfer ability. Chinese performs better than Arabic and Russian. The possible reasons are as follows: (1) Both Arabic and Russian are morphologically rich whereas Chinese is not. Morphological variation makes finding aligned ngrams in the forward pass harder, with a negative impact on performance; (2) To prevent bad tokenization for Chinese, we allow all ngrams (unlimited-length combination of continuous characters) in a verse to be candidates in the forward pass. This setting gives ngrams more freedom and thus better results are expected. For the three low-resource languages, we find that they diverge morphologically and typologically

from most high-resource languages. Apinayé and Mündü seem to frequently use several consecutive whitespace-tokenized syllables to express a single concept, which makes finding the correct alignments much harder. Salt-Yui, on the other hand, seems to be highly ambiguous because the writing does not reflect its contrastive tones (Irwin, 1974). We hypothesize such ambiguity can negatively influence performance. See §F for an analysis of the factors that can influence transfer performance.

## 6 Conclusion

In this work, we present the multilingual graphs ColexNet and ColexNet+, based on colexifications extracted from a highly parallel corpus. Comparing with CLICS, we show that we identify many gold-standard patterns in ColexNet. In addition, we analyze the structure of ColexNet and show it nearly forms a scale-free graph, with many communities of semantically related concepts. Most importantly, we contribute to crosslingual transfer learning by inducing multilingual embeddings $\overrightarrow{\text{ColexNet+}}$ that are learned on ColexNet+. Our experiments indicate that $\overrightarrow{\text{ColexNet+}}$ largely represents concepts across languages in the same semantic space. We show that $\overrightarrow{\text{ColexNet+}}$ outperforms several approaches, including multilingual embeddings and pretrained models, on three downstream tasks. This indicates that embeddings learned from colexification graphs improve crosslingual transfer, especially for low-resource languages for which it is often infeasible to pretrain good models. Finally, our embeddings exhibit robust transfer performance across many different source languages.

## Limitations

Theoretically, one could identify, explore, and analyze colexification patterns from any parallel corpora and construct graphs of colexifications using the methods proposed in this paper. We use the PBC, a genre-specific parallel corpus in this work, which can limit some of the concepts to religions. Nevertheless, the goal of this work is to explore colexification patterns in as many languages as possible, including a lot of low-resource languages, without relying on any external resources. Therefore, the PBC corpus is a good fit for us.

We conduct extensive experiments to verify the crosslingual transfer capability of the multilingual embeddings learned on ColexNet+. However, some experiments are in-domain (the evaluation tasks are still related to the Bible), e.g., verse retrieval and verse classification. The major reason is that we want to test the embedding's performance on all our supported languages. Unfortunately, as far as we know, evaluation datasets that cover such a wide range of languages, including low-resource languages, are missing in the community. Some datasets, for example, Tatoeba[4], support hundreds of languages but contain many concepts, e.g., pizza, that do not occur in the Bible. Therefore, we do not evaluate our embeddings on those datasets.

## Acknowledgement

We would like to thank Verena Blaschke for constructive discussions and Yuxuan Zahed for the implementation of the online demo of ColexNet and ColexNet+. This work was funded by the European Research Council (grant #740516).

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

# A Choice of hyperparameters and discussion

## A.1 Forward/backward pass

Two hyperparameters in the forward pass and backward pass when searching for the colexification patterns can influence the results, i.e., (1) the maximum number of iterations $M$ for a given concept in each language and (2) the threshold $\alpha$ for the minimum cumulative coverage of the set of identified ngrams. We set $M = 3$ and $\alpha = .9$ as default values for all involved computations. We are different with Conceptualizer (Liu et al., 2023) in the setting of $M$. Conceptualizer sets $M$ to 5 whereas we set it to 3. The major reasons are as follows. We are searching for colexification patterns with high accuracy. This requires us to identify the target-language ngrams that instantiate the colexifications with high certainty. Based on empirical explorations, we find that when $M$ is large (e.g., $> 3$), we can include less accurate or even unrelated ngrams (because those ngrams are rare and occur in the same verse where the concept occurs, which is also discussed by Liu et al. (2023)). By setting $M = 3$ in the forward pass, we will be more confident that the identified target-language ngrams are highly correlated with the concept and this setting achieves the best performance for a few examples in our manual inspection. As for the minimum cumulative coverage threshold $\alpha$, we directly follow the setting in Conceptualizer, i.e., 0.9, to ensure that the forward pass and backward pass find enough ngrams/concepts while guaranteeing the quality of the associations.

## A.2 ColexNet/ColexNet+ construction

In the construction of ColexNet and ColexNet+, we have an important hyperparameter $\lambda$: the minimum

| $\lambda$ | #nodes | #edges | degree | #components |
|---|---|---|---|---|
| 1 | 5870 | 1000937 | 170.61 | 1 |
| 5 | 4028 | 122798 | 30.48 | 1 |
| 10 | 3562 | 58031 | 16.29 | 1 |
| 20 | 3133 | 30175 | 9.63 | 2 |
| 50 | 2591 | 13607 | 5.25 | 9 |
| 100 | 2221 | 7634 | 3.44 | 60 |

Table 4: Basic statistics of ColexNet under different thresholds $\lambda$. We report the number of nodes (#nodes), the number of edges (#edges), the average degree per node (degree), and the number of connected components (#components).

| $\lambda$ | #nodes | #edges | degree |
|---|---|---|---|
| 1 | 3,613,546 | 15,251,571 | 4.22 |
| 5 | 3,611,704 | 12,197,241 | 3.37 |
| 10 | 3,611,238 | 11,227,402 | 3.11 |
| 20 | 3,610,809 | 10,369,418 | 2.87 |
| 50 | 3,610,267 | 9,235,395 | 2.56 |
| 100 | 3,609,897 | 8,314,760 | 2.30 |

Table 5: Basic statistics of ColexNet+ under different thresholds $\lambda$. We report the number of nodes (#nodes), the number of edges (#edges), and the average degree per node (degree).

number of languages for a colexification edge to be included. As shown in Table 4, different $\lambda$ can influence the number of nodes and edges in ColexNet as well as the number of connected components. It is clear that both #edges and degree decrease dramatically from $\lambda = 1$ to 5, which might indicate: (1) increasing $\lambda$ decreases the number of incorrectly identified colexification patterns (e.g., due to verse-level misalignment); (2) some colexification patterns might be specific to very few languages. Because of many plausible incorrect edges between concepts, when $\lambda = 1$, 5 or 10, ColexNet forms a large connected graph. When $\lambda$ is larger (e.g., 50 or 100), the graph is no longer connected because many less reliable edges are removed from it.

The influence of $\lambda$ also apply to ColexNet+, since edges being removed in ColexNet also impact ColexNet+: pairs of edges that are expanded from removed edges from ColexNet are then not included in ColexNet+. We show the number of nodes and edges as well as the average degree in ColexNet+ under different $\lambda$ in Table 5. The changes in degree with the increase of $\lambda$ are not as prominent as in ColexNet (shown in Table 4). This is mainly because the number of nodes in ColexNet+ is far more than that in ColexNet. Most of the nodes are only associated with around 3 other nodes in ColexNet+, which indicates that many ngrams from target languages colexify about three

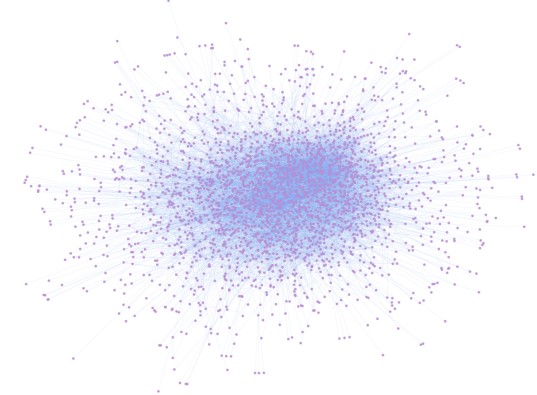

Figure 4: Visualization of the largest community which contains 2,581 nodes out of 2,591 nodes in ColexNet. Each node is a concept and each edge indicates that two concepts are colexified in at least 50 languages.

concepts, because most of the nodes in ColexNet+ belongs to the ngram nodes. Each concept, however, can be frequently associated with more than 3 concepts in ColexNet, as we noticed the average degree of ColexNet ($\lambda = 50$) is around 5.

The number of nodes and edges also influences the random walks which we used for sampling, thus influencing the quality of multilingual embeddings trained on ColexNet+ using Node2Vec (Grover and Leskovec, 2016). Therefore, we conduct experiments using embeddings trained on ColexNet+ under different $\lambda$. Same as §4, we conduct experiments on roundtrip translation, verse retrieval, and verse classification tasks. For roundtrip translation, we again set $l_0 =$ English and use 2,221 words that occur in all embeddings as the start points. For verse retrieval (resp. classification), we also use English as the query (resp. train) language, and report top-$k$ accuracy (resp. macro $F_1$ score), averaged over all languages. Results are shown in Table 6.

We see there are different trends between the changes in $\lambda$ and changes in performance for different tasks: (1) the performance of roundtrip translation is positively correlated with $\lambda$ and the best result is achieved when $\lambda = 100$; (2) the performance of verse retrieval is also positively correlated with $\lambda$; (3) the performance of verse classification is generally negatively correlated with $\lambda$ where the best result is achieved when $\lambda = 1$. Those trends can be explained as follows. Roundtrip translation and verse retrieval, compared with verse classification, require better alignment quality among concepts and ngrams. When $\lambda$ is small, some incorrect edges are included in the graph. These edges induce

| $\lambda$ | Roundtrip translation | | | verse retrieval | | | verse classification |
|---|---|---|---|---|---|---|---|
| | top-1 | top-5 | top-10 | top-1 | top-5 | top-10 | |
| 1 | 0.29 | 0.66 | 0.78 | 0.51 | 0.68 | 0.74 | **0.51** |
| 5 | 0.34 | 0.71 | 0.82 | 0.58 | 0.74 | 0.79 | 0.49 |
| 10 | 0.36 | 0.74 | 0.84 | 0.59 | 0.75 | 0.81 | 0.48 |
| 20 | 0.38 | 0.77 | 0.87 | 0.63 | 0.78 | 0.83 | 0.48 |
| 50 | 0.40 | 0.81 | 0.91 | 0.65 | 0.80 | 0.84 | 0.49 |
| 100 | **0.42** | **0.84** | **0.93** | **0.66** | **0.81** | **0.85** | 0.45 |

Table 6: Results of multilingual embeddings trained on ColexNet+ under different $\lambda$. Each number for roundtrip translation (top-$k$ accuracy, $k = [1, 5, 10]$) is the average of 10 runs with 3 randomly selected intermediate languages. Each number in verse retrieval (top-$k$ accuracy, $k = [1, 5, 10]$) and verse classification (macro $F_1$) is the average over **all** available languages. **Bold** (underlined): best (second-best) result per column.

noises for sampling, therefore slightly noisy embeddings are obtained, negatively influencing the performance. As for verse classification, the results suggest that when we have fewer out-of-vocabulary ngrams in the embeddings (higher $\lambda$ induces fewer ngrams in ColexNet+), slightly better performance is achieved. Moreover, $\lambda$ seems to have a more obvious impact on roundtrip translation and verse retrieval than on verse classification. In summary, the results verify our choice of $\lambda$ =50, a relatively large number, in the main content of this paper, as it offers very competitive results compared to other choices while not losing many interesting patterns.

## B   Investigation of identified colexifications

In §4.2, we show that we identify many ground-truth colexification patterns compared with CLICS. However, there are quite a few colexification patterns that are not present in CLICS. Therefore, we conduct a qualitative investigation on those colexification patterns. We classify each pattern (an colexification edge in ColexNet between two concepts) as one of the following categories: (1) full colexification (2) partial colexification and (3) incorrect colexification.

**Full colexification.**   Full colexification indicates that a word in a language directly colexifies two concepts. We list 4 examples of colexifications not included in CLICS but verified by us. An obvious example is that <ground> and <land> are colexified in many languages, e.g., through 土地 in Japanese (jpn) and 大地 in Chinese (zho). <early> and <tomorrow> are frequently colexified in Turkic languages, e.g.,

- Southern Altai (alt): $эртен

- Bashkir (bak): $иртэн$

- Kyrgyz (kir): $эртен

- Nogai (nog): $эртен

Another interesting example is that <love> and <wish> are frequently colexified, e.g., in Min Nan Chinese (nan) through the word $ài (character: 愛), as the character means both <love> and <wish>. Lastly, in Western Frisian (fry), <dragon> and <snake> are colexified through the word $draek$, for which we manually verify in PBC. It is worth noting that there is another word slang which denotes <snake> in Western Frisian.

**Partial colexification.**   Partial colexification denotes the pattern that does not involve an entire word, but rather part of it. The part can be a shared root or a shared element in a compound. Since our algorithm works on the character ngrams within a word, we find many partial colexification patterns. For example, <stand> and <build> are colexified in Kazakh (kaz) through $тұр. In Kazakh, <stand> is expressed by the word тұру while <build> by тұрғызу. тұрғызу is actually derived from the root тұр- plus a causative suffix -ғыз, so that тұрғыз- means 'to make something stand', thus meaning 'build, erect'. Such partial colexifications through a root can even include more than two concepts, e.g., <morning>, <early>, and <next> in many Turkic languages:

- Turkmen (tuk):
  $ertesi → <next>
  $erte → <morning>

- Turkish (tur):
  $ertesi → <next>
  $erte → <morning> <early>

- Uyghur (uig):
  $әтиси$ → <next>
  $әтигән → <morning>
  $әти → <early>

| types | incorrect colex. | languages | ngrams | context |
|---|---|---|---|---|
| **co-occurrence** | <four> <twenty> | cat | `$quatre$`, `$vint` | #66004004: ... **vint**-i-**quatre** setials més ... |
| | <left> <right> | nog | `$онъ$`, `$сол$` | #01048013: ... онъ колы ... сол ... |
| | <want> <know> | nds | `$weete$`, `$well$` | #46011003: Oba etj **well** , daut jie **weete** ... |
| **free translation** | <man> <answer> | cat | `$contest` | #40012048: Però ell va **contest**ar ... |
| | <hundred> <thousand> | cmn | 十万 | #13005021: ... 以及人口十万... |

Table 7: Examples of incorrectly identified colexifications in ColexNet.

Some concepts may be expressed using multiple lexemes, forming a compound, and a part of the compound may also occur in the expression of a different concept. Note that, in some languages (such as German), a compound is written together without any space in between, whereas in some other languages (such as English), there is a space between each part. In either case, these are considered as compounds, since all the separate elements together constitute one concept. This cannot be confused with co-occurrence, where the two concepts themselves co-occur. For example, in Tatar (tat), two colors: <purple> and <scarlet> are partially colexified through $кызыл$, because <purple> is куе кызыл (literally 'thick red'), which contains a part кызыл meaning 'red, scarlet'. Such partial relation also frequently exists in numbers. For example, `empat belas` (resp. 十四), which means 14, and `empat puluh` (resp. 四十), which means 40, are partially colexified in Indonesian (ind) (resp. Chinese (zho)), as `empat` (resp. 四) means 4. Some languages, e.g., Chinese and German, construct compounds without inserting blanks between each lexeme, so we also observe many partial colexifications in Chinese and Germanic languages, e.g., :

- Chinese (cmn):
  震 → <tremble>
  地震 → <earthquake>

- Bavarian (bar):
  `bibn` → <tremble>
  `$erdbibn$` → <earthquake>

- German (deu):
  `$beben` → <tremble>
  `$erdbeben$` → <earthquake>

In summary, many identified colexification patterns in ColexNet belong to this category, which is the reason why we found many patterns that do not exist in CLICS, since CLICS only includes full colexification patterns.

**Incorrect colexification.** As an automatic statistical method, the results are not immune to errors. Typically, we find the incorrectly identified colexifications are mainly due to two reasons: (1) **co-occurrence** and (2) **free translation**. We list some incorrectly identified colexifications in Table 7. Co-occurrence denotes that two particular concepts tend to co-occur very often so that the algorithm wrongly establishes connections between the concept. For example, we found <four> and <twenty> are associated in Catalan (cat) because the ngrams `$quatre$`, `$vint` which refer to the two concepts respectively co-occur very frequently in PBC. Similarly, <left> and <right> for Nogai (nog), and <want> and <know> for Low German (nds) also belong to this type of error. Free translation means that the translation is not done word by word so that the corresponding word for a specific concept does not occur in the same sentence. In this case, the algorithm has no chance of finding the corresponding ngram, which ideally would align with the intended concept. Free translation is very common in the Bible because of its religious textbook nature. For example, in Catalan (cat), the English verse #40012048 starts with "But to the **man** who told him" but the Catalan translation starts with "Però ell va **contest**ar al qui deia això", which means "*But he **answer**ed the one who said this*", where the concept <man> does not occur in Catalan and the concept <answer> does not occur in English verse. Similarly, Chinese word 十万 means one hundred thousand, i.e., 100,000 (with 十 being 10 and 万 being 10,000). As the formation of the number expression in Chinese is different from its English counterpart, the algorithm wrongly associates <hundred> and <thousand>.

## C   Communities of ColexNet

There are 288 communities in total detected in ColexNet ($\lambda = 50$) by Louvain community detection algorithm (Blondel et al., 2008). Two important hyperparameters, i.e., resolution and random seed are set to 0.1 and 114514 respectively. As

mentioned in §5, each community is a cluster of concepts that are semantically related to each other. We create a demonstration website to show the subnetworks of each concept and the community figures.[5] For illustration purposes, we randomly select 15 communities that have more than 10 nodes for illustration in this paper. See Visualizations of those communities in Figure 5.

## D  Influence of language families & areas

We create subnetworks specific to each language family and to each area. We consider six language families that have more than 50 languages in PBC: Austronesian (**aust**), Atlantic-Congo (**atla**), Indo-European (**indo**), Nuclear Trans New Guinea (**nucl**), Otomanguean (**otom**) and Sino-Tibetan (**sino**). We consider five areas: South America (**SA**), North America (**NA**), **Eurasia**, **Africa** and **Papunesia**. We only keep the edges in ColexNet that occur in each language family (resp. area) for the subnetwork of each language family (resp. area). To quantify agreement of community structure, we use adjusted rand index (ARI) (Hubert and Arabie, 1985; Steinley, 2004), similar to (Jackson et al., 2019). We also compute ARI between ColexNet and each subnetwork. Figures 6 and 7 show pairwise ARI for language families and areas. It is clear that any language family subnetwork cannot represent the global colexification patterns encoded in ColexNet, since no family's ARI with ColexNet is high. In addition, no two language families have a similar community structure according to ARI: for the pair with the highest ARI, **atla**-**aust**, ARI = 0.5. In comparison, area-specific subnetworks generally have larger pairwise ARIs. The two areas **Africa** and **Papunesia** have a very high ARI of 0.76 and also high ARIs with ColexNet (0.78 and 0.80). This can be explained by (1) there are many languages in those two areas so there are more possible colexifications included in the subnetworks and (2) the diversity (in terms of colexification) of languages spoken in these two areas is high. In summary, relatively low ARIs between families and areas also suggest many colexification patterns are only specific to a small group of languages (either in a specific language family or in an area).

[5] https://conceptexplorer.cis.lmu.de/

## E  English-centric transfer learning

We have shown the English-centric transfer performance of verse retrieval and verse classification averaged over languages in Table 2. We believe it is also important to have a fine-grained view of the results for individual languages, to better understand the crosslingual transfer capability of $\overrightarrow{\text{ColexNet+}}$. Therefore, we show the transfer performance (sentence retrieval and sentence classification) of each individual language clustered by its corresponding language family in Figure 8. Globally, we see that results not only vary across language families but also vary within each language family. English

We find in Figure 8 (top) that, though a top-10 accuracy higher than 0.5 is achieved for all languages, the average retrieval accuracy in the Indo-European language family is slightly better than in other families which have many languages (e.g., Sino-Tibetan or Otomanguean language family). We speculate this is probably because other Indo-European languages can learn more accurate alignments as our source language is English which also belongs to the same family. Better alignments influence the quality of embeddings in that language, therefore having an impact on the transfer performance.

The trend in classification as shown in Figure 8 (bottom), however, is slightly different: average $F_1$ remains stable at around 0.5 for almost all language families, with less variance in each language family. This is evidence for our conjecture that classification is a less difficult task: apparently, good performance can be obtained if only words referring to the most important concepts that are highly associated with specific classes are aligned well.

In summary, good performance indicates that $\overrightarrow{\text{ColexNet+}}$ assigns similar representations to ngrams that refer to the same concept, thus improving crosslingual transfer.

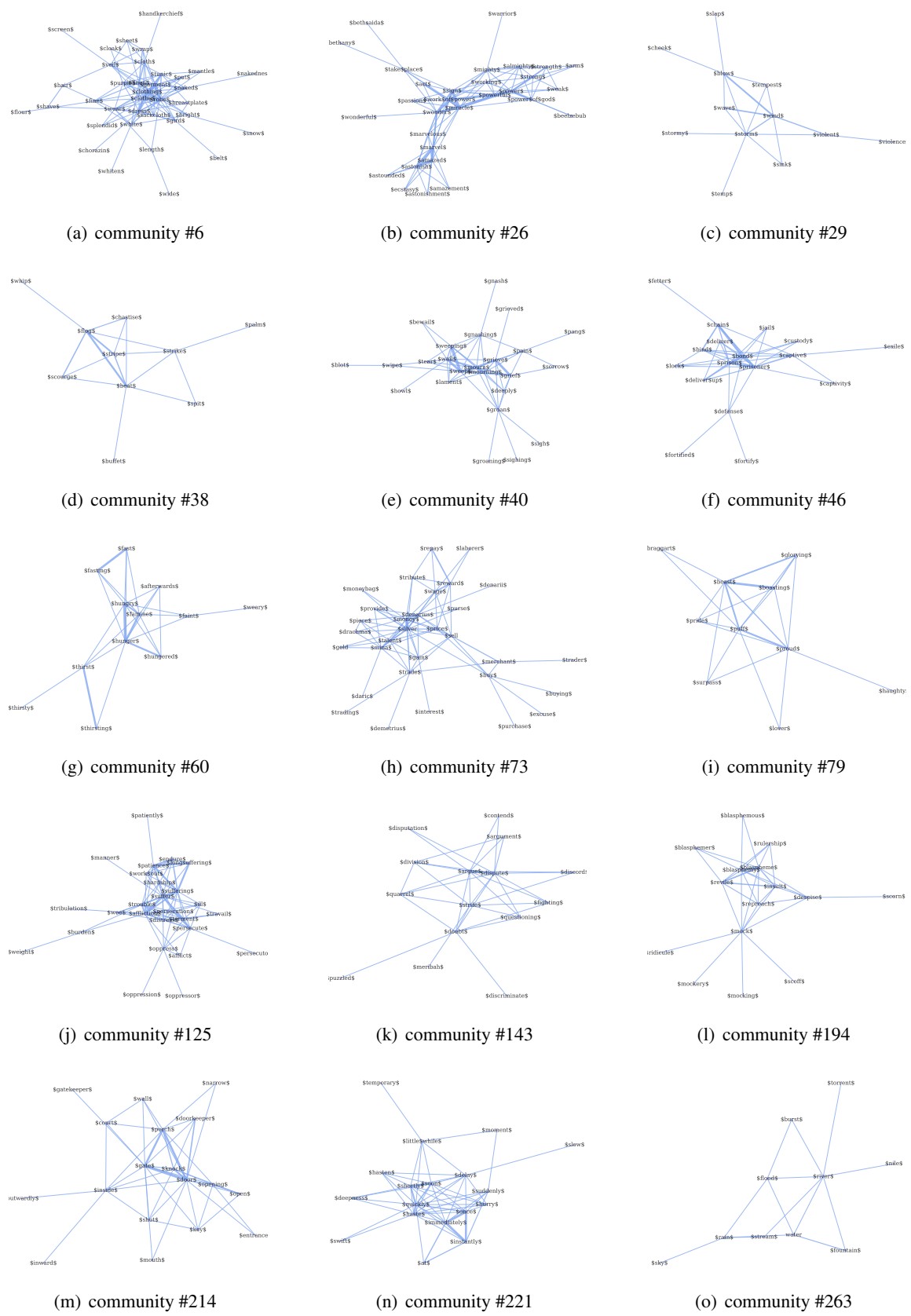

Figure 5: Visualizations of 15 randomly selected communities that have more than 10 nodes from 288 communities detected in ColexNet. Each community forms a cluster of concepts that are semantically related to each other. E.g., community #60 is related to the concept <hunger>; and community #73 is related to the concept <money>.

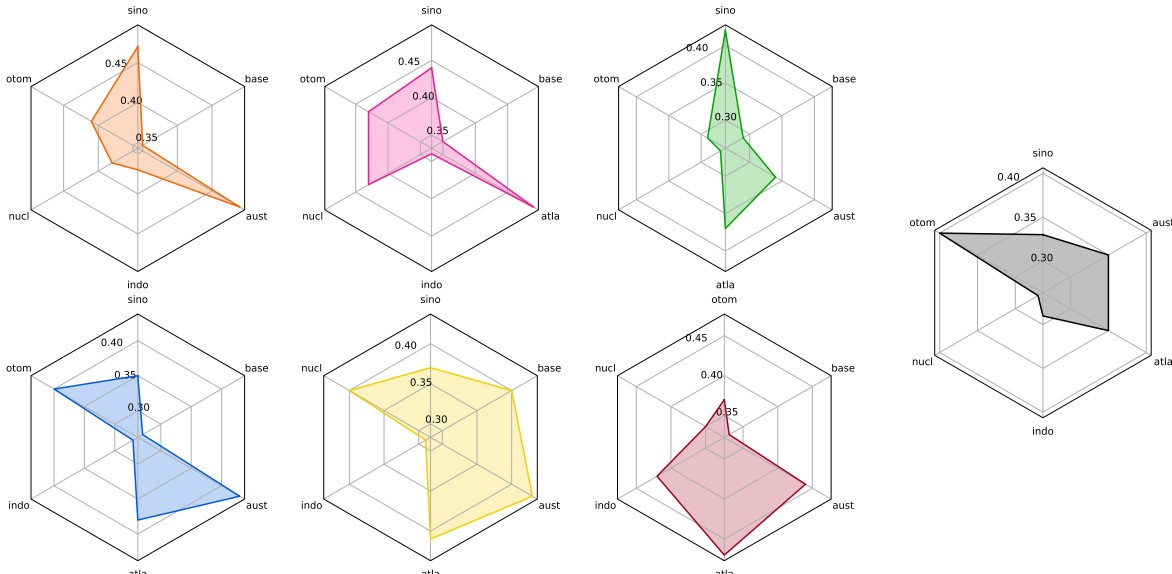

Figure 6: Pairwise ARIs between language family-specific subnetworks. Each subfigure contains pairwise ARIs between one family (indicated by the color: **atla**, **aust**, **indo**, **nucl**, **otom**, **sino**, **base**) and all other families (indicated on the edges). The ARIs are computed by averaging the results of 50 runs using the Louvain algorithm with different random states. Pairs of the same family, e.g., **indo-indo**, are not shown because the ARI will always be 1 in such cases. **base** is the graph including all edges, i.e., ColexNet. Note that the scale is adjusted for each family individually.

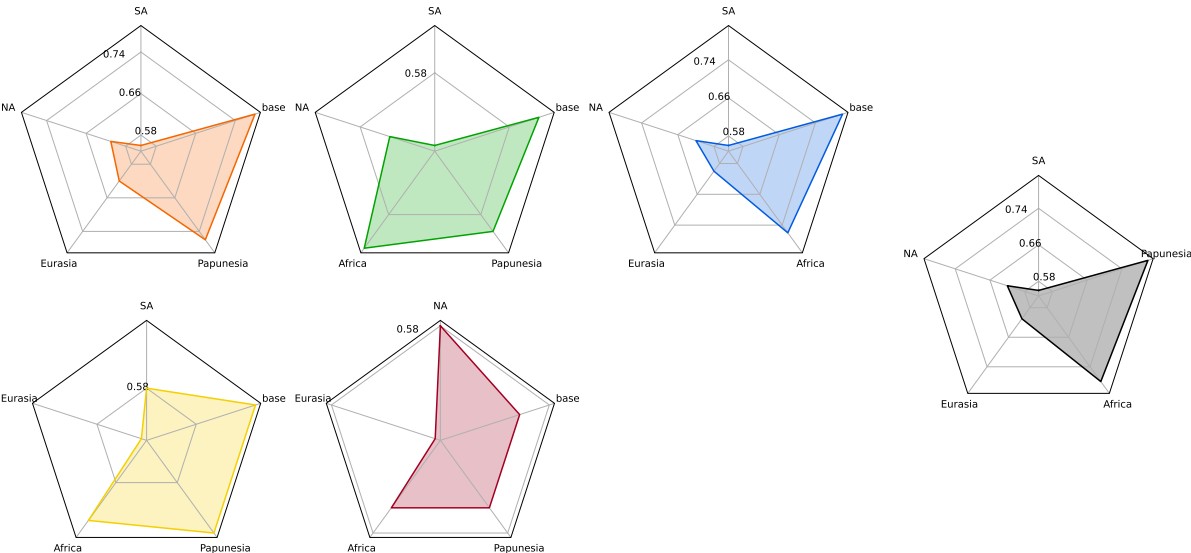

Figure 7: Pairwise ARIs between area-specific subnetworks. Each subfigure contains pairwise ARIs between one area (indicated by the color: **Africa**, **Eurasia**, **Papunesia**, **NA**, **SA**, **base**) and all other areas (indicated on the edges). The ARIs are computed by averaging the results of 50 runs using the Louvain algorithm with different random states. Pairs of the same area, e.g., **Africa-Africa**, are not shown because the ARI will always be 1 in such cases. **base** is the graph including all edges, i.e., ColexNet. Note that the scale is adjusted for each area individually.

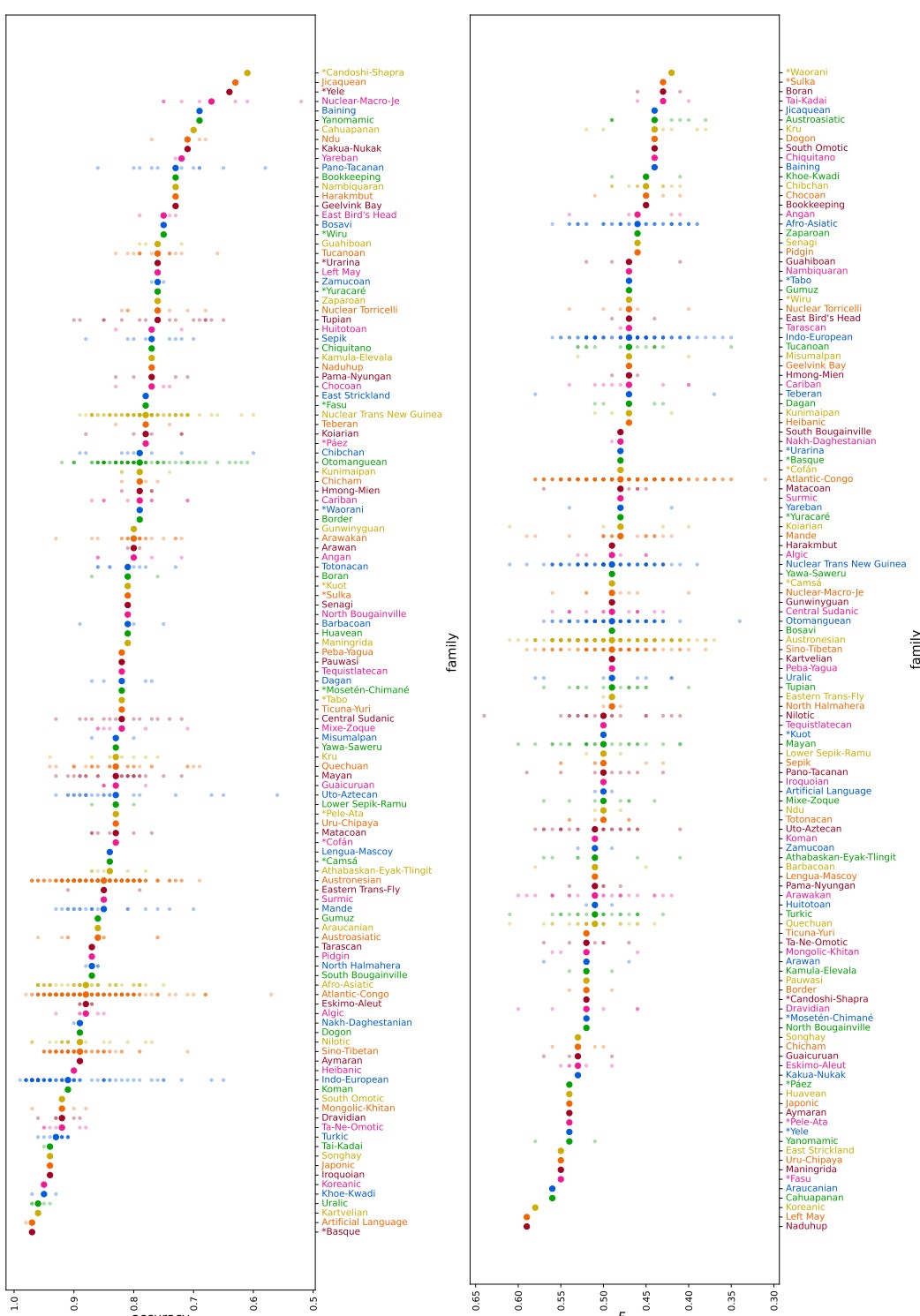

Figure 8: Top-10 accuracy of verse retrieval (top) and $F_1$ for verse classification (bottom) by language family for $\overleftrightarrow{\text{ColexNet+}}$. Each small dot represents a language, each large dot an average per family. Families are color-coded for better readability and sorted by accuracy/$F_1$. Asterisk (*) before name: language isolate. Most results are good (average accuracy higher than 0.8 and average $F_1$ higher than 0.5 for half of the language families), but transfer performance varies across language families.

## F  Beyond English-centric transfer

We show the complete transfer performance by using any language as train/query language (1,245 languages in total, as we filtered some languages which have a very small size of train or test set). The results are shown in Table 10, 11, 12, 13, 14.

We hypothesize that the quality of identified colexification can influence the transfer learning performance. Some languages, because their morphology, typology, or conceptualization are very different from other languages, might pose difficulties in finding reliable colexification patterns, thus being detrimental to crosslingual transfer. To this end, we compute the average colexification patterns per ngram (avg_colex) for each language. That is, for language $l$, we compute the average number of neighbors of an ngram in ColexNet+. The neighbors of an ngram node are concept nodes, which indicates the concepts that this ngram can refer to. The higher the avg_colex is for a language, the more polysemous or ambiguous the ngrams tend to be. Of course, the extracted colexification patterns are not always correct due to verse-level misalignment, free translation, or some language-specific properties like morphology. Therefore the metric avg_colex can, to some degree, indicate the level of difficulty to find correct alignments.

We list the number of target-language ngrams in ColexNet+ (#ngrams) as well as avg_colex for the languages we show in §5.2: Arabic (arb), Russian (rus), Chinese (zho), Apinayé (apn), Mündü (muh), Salt-Yui (sll) as well as the average over all languages in Table 8. Three high-resource languages, which are typologically and morphologically different from each other, show similar trends in their statistics: more ngrams are included in ColexNet+ while avg_colex is less than the average. This might indicate that the languages are less ambiguous and the colexifications extracted are mostly reliable, which explains good crosslingual performance when they are used as the train/query languages. On the contrary, the three worst-performing languages have exactly the inverse trend, which indicates it is harder to identify reliable colexifications, thus the performance is bad when they are served as the source languages.

To further test our hypothesis, we compute the Pearson correlation between the performance (classification $F_1$ score and retrieval accuracy) and avg_colex. The results are shown in Table 9. It is evident that #ngrams is weakly positively cor-

| language | #ngrams | avg_colex |
|---|---|---|
| arb | 4,107 | 1.84 |
| rus | 3,574 | 2.21 |
| zho | 3,659 | 2.07 |
| apn | 2,119 | 2.91 |
| muh | 1,408 | 4.00 |
| sll | 2,118 | 2.93 |
| avg. | 2,702 | 2.64 |

Table 8:   Number of target-language ngrams in ColexNet+ (#ngrams) and the average number of colexified concepts per ngram (avg_colex) for Arabic (arb), Russian (rus), Chinese (zho), Apinayé (apn), Mündü (muh), Salt-Yui (sll) as well as the average over all languages. We see that the lower three worst performing languages have fewer #ngrams but larger avg_colex than the average statistics over all languages.

|  | c | r1 | r5 | r10 |
|---|---|---|---|---|
| #ngrams | 0.20 | 0.28 | 0.25 | 0.24 |
| avg_colex | -0.18 | -0.25 | -0.21 | -0.19 |

Table 9:   Pearson correlations between #ngrams/ avg_colex and the transfer performance (c: classification $F_1$ score, r1: retrieval top-1 accuracy, r5: retrieval top-5 accuracy, r10: retrieval top-10 accuracy). All values are statistically significant under $p = 0.01$.

related with the performance while avg_colex is negatively correlated. However, it is important to note that the correlation is not high: there are quite a few languages that have small #ngrams but large avg_colex perform quite well when they are used as the source languages for large-scale transfer. For example, Bislama (bis), whose #ngrams is only 1,202 but avg_colex is 4.81, achieves good performance: 0.41, 0.46, 0.66, 0.73 for classification, retrieval top-1, top-5, and top-10 respectively. We speculate this is because Bislama is highly influenced by English (Tryon, 1987), therefore the patterns extracted are reliable since the concepts are represented in English lemmata. We leave the further exploration of finding reliable colexifications from a parallel corpus for future research.

To sum up, the quality of the colexification patterns extracted for a language is closely related to the transfer performance when it is served as the train/query language. Due to various language-specific properties, the model can have difficulties in inducing reliable colexification patterns.

| language | classification | retrieval | | | language | classification | retrieval | | | language | classification | retrieval | | |
|---|---|---|---|---|---|---|---|---|---|---|---|---|---|---|
| | | top-1 | top-5 | top-10 | | | top-1 | top-5 | top-10 | | | top-1 | top-5 | top-10 |
| aai | 0.46 | 0.47 | 0.65 | 0.72 | aak | 0.42 | 0.39 | 0.57 | 0.64 | aau | 0.45 | 0.47 | 0.67 | 0.74 |
| aaz | 0.40 | 0.39 | 0.58 | 0.65 | abt | 0.42 | 0.32 | 0.50 | 0.57 | abx | 0.44 | 0.52 | 0.70 | 0.76 |
| aby | 0.40 | 0.30 | 0.49 | 0.57 | acd | 0.44 | 0.39 | 0.58 | 0.66 | ace | 0.44 | 0.51 | 0.69 | 0.75 |
| acf | 0.43 | 0.52 | 0.71 | 0.77 | ach | 0.48 | 0.53 | 0.71 | 0.77 | acn | 0.44 | 0.51 | 0.70 | 0.76 |
| acr | 0.42 | 0.47 | 0.65 | 0.72 | acu | 0.43 | 0.39 | 0.58 | 0.66 | ade | 0.46 | 0.45 | 0.64 | 0.71 |
| adh | 0.47 | 0.53 | 0.71 | 0.77 | adi | 0.45 | 0.48 | 0.66 | 0.72 | adj | 0.43 | 0.42 | 0.61 | 0.68 |
| adl | 0.44 | 0.44 | 0.63 | 0.70 | aeb | 0.46 | 0.54 | 0.72 | 0.78 | aeu | 0.44 | 0.44 | 0.63 | 0.71 |
| aey | 0.46 | 0.47 | 0.65 | 0.72 | afr | 0.47 | 0.58 | 0.74 | 0.79 | agd | 0.43 | 0.38 | 0.58 | 0.66 |
| agg | 0.44 | 0.43 | 0.62 | 0.69 | agm | 0.44 | 0.39 | 0.57 | 0.64 | agn | 0.44 | 0.50 | 0.68 | 0.75 |
| agr | 0.41 | 0.43 | 0.62 | 0.69 | agt | 0.43 | 0.38 | 0.56 | 0.63 | agu | 0.42 | 0.40 | 0.59 | 0.66 |
| agw | 0.46 | 0.41 | 0.60 | 0.68 | ahk | 0.44 | 0.38 | 0.57 | 0.65 | aia | 0.42 | 0.44 | 0.63 | 0.71 |
| aii | 0.43 | 0.55 | 0.71 | 0.77 | aim | 0.42 | 0.41 | 0.61 | 0.68 | aji | 0.44 | 0.48 | 0.67 | 0.73 |
| ajz | 0.46 | 0.52 | 0.70 | 0.77 | akb | 0.43 | 0.51 | 0.70 | 0.76 | ake | 0.47 | 0.48 | 0.66 | 0.73 |
| akh | 0.46 | 0.31 | 0.49 | 0.58 | ald | 0.41 | 0.49 | 0.68 | 0.75 | alj | 0.45 | 0.46 | 0.65 | 0.73 |
| aln | 0.47 | 0.56 | 0.72 | 0.78 | alp | 0.41 | 0.41 | 0.60 | 0.68 | alq | 0.44 | 0.51 | 0.69 | 0.76 |
| alt | 0.47 | 0.58 | 0.75 | 0.81 | alz | 0.45 | 0.52 | 0.69 | 0.75 | ame | 0.40 | 0.40 | 0.58 | 0.66 |
| amf | 0.47 | 0.55 | 0.72 | 0.78 | amh | 0.44 | 0.55 | 0.73 | 0.78 | amk | 0.42 | 0.48 | 0.66 | 0.73 |
| amm | 0.43 | 0.39 | 0.59 | 0.66 | amn | 0.45 | 0.40 | 0.59 | 0.67 | amp | 0.40 | 0.34 | 0.52 | 0.60 |
| amr | 0.41 | 0.33 | 0.52 | 0.59 | amu | 0.42 | 0.46 | 0.64 | 0.71 | ann | 0.44 | 0.44 | 0.63 | 0.70 |
| anv | 0.44 | 0.42 | 0.61 | 0.68 | aoj | 0.45 | 0.41 | 0.60 | 0.67 | aom | 0.41 | 0.38 | 0.57 | 0.64 |
| aon | 0.42 | 0.39 | 0.58 | 0.66 | aoz | 0.51 | 0.52 | 0.70 | 0.76 | apb | 0.43 | 0.47 | 0.65 | 0.72 |
| ape | 0.39 | 0.40 | 0.60 | 0.67 | apn | 0.38 | 0.21 | 0.38 | 0.46 | apr | 0.44 | 0.40 | 0.60 | 0.68 |
| apt | 0.45 | 0.52 | 0.69 | 0.76 | apu | 0.45 | 0.44 | 0.62 | 0.69 | apw | 0.45 | 0.47 | 0.65 | 0.72 |
| apy | 0.42 | 0.35 | 0.53 | 0.61 | apz | 0.45 | 0.35 | 0.54 | 0.61 | arb | 0.47 | 0.56 | 0.72 | 0.78 |
| are | 0.41 | 0.41 | 0.59 | 0.67 | arl | 0.44 | 0.36 | 0.55 | 0.63 | arn | 0.41 | 0.48 | 0.67 | 0.73 |
| arz | 0.46 | 0.53 | 0.70 | 0.76 | asg | 0.42 | 0.41 | 0.60 | 0.67 | aso | 0.45 | 0.38 | 0.57 | 0.64 |
| ata | 0.46 | 0.44 | 0.64 | 0.71 | atb | 0.44 | 0.53 | 0.71 | 0.77 | atd | 0.45 | 0.44 | 0.64 | 0.71 |
| atg | 0.46 | 0.45 | 0.64 | 0.71 | att | 0.43 | 0.47 | 0.66 | 0.73 | auc | 0.45 | 0.38 | 0.57 | 0.64 |
| auy | 0.42 | 0.37 | 0.56 | 0.64 | ava | 0.46 | 0.52 | 0.70 | 0.76 | avt | 0.44 | 0.27 | 0.45 | 0.53 |
| avu | 0.42 | 0.29 | 0.48 | 0.56 | awa | 0.43 | 0.50 | 0.68 | 0.75 | awb | 0.41 | 0.42 | 0.60 | 0.67 |
| awi | 0.44 | 0.34 | 0.53 | 0.61 | ayo | 0.42 | 0.36 | 0.56 | 0.63 | ayr | 0.42 | 0.47 | 0.65 | 0.72 |
| aze | 0.47 | 0.53 | 0.70 | 0.76 | azg | 0.42 | 0.41 | 0.59 | 0.67 | azz | 0.45 | 0.45 | 0.64 | 0.71 |
| bak | 0.45 | 0.55 | 0.72 | 0.78 | bam | 0.47 | 0.57 | 0.74 | 0.80 | ban | 0.46 | 0.52 | 0.70 | 0.76 |
| bao | 0.47 | 0.39 | 0.58 | 0.66 | bar | 0.48 | 0.47 | 0.64 | 0.71 | bav | 0.47 | 0.42 | 0.62 | 0.69 |
| bba | 0.45 | 0.51 | 0.69 | 0.76 | bbb | 0.42 | 0.34 | 0.53 | 0.60 | bbj | 0.47 | 0.45 | 0.64 | 0.71 |
| bbr | 0.43 | 0.36 | 0.55 | 0.62 | bch | 0.44 | 0.44 | 0.64 | 0.71 | bci | 0.48 | 0.42 | 0.61 | 0.68 |
| bcl | 0.49 | 0.58 | 0.75 | 0.81 | bcw | 0.43 | 0.42 | 0.60 | 0.68 | bdd | 0.43 | 0.44 | 0.63 | 0.70 |
| bdh | 0.36 | 0.34 | 0.53 | 0.61 | bef | 0.46 | 0.34 | 0.53 | 0.60 | bel | 0.43 | 0.61 | 0.77 | 0.82 |
| bem | 0.45 | 0.52 | 0.70 | 0.76 | ben | 0.49 | 0.52 | 0.69 | 0.76 | beq | 0.47 | 0.56 | 0.73 | 0.79 |
| bex | 0.41 | 0.38 | 0.58 | 0.65 | bfd | 0.47 | 0.47 | 0.65 | 0.72 | bfo | 0.40 | 0.49 | 0.67 | 0.74 |
| bgr | 0.42 | 0.52 | 0.70 | 0.77 | bgs | 0.44 | 0.49 | 0.68 | 0.75 | bgz | 0.45 | 0.51 | 0.69 | 0.75 |
| bhl | 0.45 | 0.30 | 0.49 | 0.57 | bhp | 0.45 | 0.49 | 0.67 | 0.74 | bib | 0.41 | 0.46 | 0.65 | 0.72 |
| big | 0.48 | 0.38 | 0.56 | 0.64 | bim | 0.46 | 0.49 | 0.68 | 0.74 | bis | 0.41 | 0.46 | 0.66 | 0.73 |
| biu | 0.47 | 0.56 | 0.73 | 0.79 | biv | 0.42 | 0.46 | 0.64 | 0.71 | bjr | 0.45 | 0.33 | 0.52 | 0.60 |
| bjv | 0.40 | 0.37 | 0.57 | 0.65 | bkd | 0.47 | 0.49 | 0.68 | 0.75 | bkq | 0.44 | 0.33 | 0.51 | 0.59 |
| bku | 0.44 | 0.45 | 0.64 | 0.71 | bkv | 0.40 | 0.47 | 0.66 | 0.73 | blh | 0.40 | 0.43 | 0.63 | 0.70 |
| blw | 0.43 | 0.45 | 0.64 | 0.71 | blz | 0.45 | 0.54 | 0.72 | 0.78 | bmb | 0.46 | 0.56 | 0.73 | 0.79 |
| bmh | 0.41 | 0.32 | 0.51 | 0.59 | bmq | 0.43 | 0.45 | 0.65 | 0.71 | bmr | 0.46 | 0.46 | 0.65 | 0.72 |
| bmu | 0.46 | 0.44 | 0.63 | 0.70 | bnj | 0.41 | 0.47 | 0.67 | 0.74 | bnp | 0.47 | 0.41 | 0.60 | 0.67 |
| boa | 0.42 | 0.35 | 0.53 | 0.61 | boj | 0.41 | 0.41 | 0.60 | 0.68 | bom | 0.43 | 0.52 | 0.70 | 0.77 |
| bon | 0.45 | 0.38 | 0.58 | 0.66 | box | 0.43 | 0.46 | 0.66 | 0.73 | bpr | 0.42 | 0.50 | 0.69 | 0.75 |
| bps | 0.43 | 0.48 | 0.66 | 0.73 | bqc | 0.42 | 0.47 | 0.65 | 0.72 | bqj | 0.47 | 0.53 | 0.71 | 0.77 |
| bqp | 0.46 | 0.50 | 0.68 | 0.74 | bru | 0.43 | 0.40 | 0.60 | 0.68 | bsc | 0.44 | 0.54 | 0.72 | 0.78 |
| bsn | 0.41 | 0.31 | 0.47 | 0.54 | bss | 0.44 | 0.50 | 0.68 | 0.74 | btd | 0.47 | 0.50 | 0.68 | 0.75 |
| bth | 0.50 | 0.57 | 0.74 | 0.80 | bto | 0.50 | 0.57 | 0.74 | 0.80 | btt | 0.45 | 0.40 | 0.60 | 0.67 |
| btx | 0.47 | 0.57 | 0.74 | 0.80 | bud | 0.44 | 0.52 | 0.70 | 0.76 | bug | 0.47 | 0.52 | 0.70 | 0.76 |
| buk | 0.45 | 0.39 | 0.59 | 0.66 | bul | 0.46 | 0.57 | 0.74 | 0.79 | bum | 0.46 | 0.44 | 0.62 | 0.69 |
| bus | 0.45 | 0.51 | 0.69 | 0.75 | bvr | 0.41 | 0.38 | 0.57 | 0.65 | bvz | 0.41 | 0.32 | 0.51 | 0.59 |
| bwq | 0.46 | 0.48 | 0.67 | 0.74 | bwu | 0.46 | 0.42 | 0.61 | 0.69 | bxr | 0.45 | 0.54 | 0.71 | 0.77 |
| byr | 0.45 | 0.45 | 0.64 | 0.71 | byx | 0.43 | 0.29 | 0.48 | 0.56 | bzd | 0.44 | 0.45 | 0.64 | 0.71 |
| bzh | 0.39 | 0.39 | 0.59 | 0.66 | bzi | 0.42 | 0.46 | 0.65 | 0.72 | bzj | 0.46 | 0.48 | 0.66 | 0.73 |
| caa | 0.45 | 0.40 | 0.60 | 0.68 | cab | 0.42 | 0.49 | 0.66 | 0.73 | cac | 0.44 | 0.37 | 0.56 | 0.64 |
| caf | 0.44 | 0.42 | 0.61 | 0.68 | cag | 0.45 | 0.45 | 0.64 | 0.71 | cak | 0.43 | 0.41 | 0.60 | 0.68 |
| cao | 0.47 | 0.40 | 0.59 | 0.66 | cap | 0.42 | 0.44 | 0.63 | 0.70 | caq | 0.45 | 0.51 | 0.69 | 0.75 |
| car | 0.43 | 0.50 | 0.68 | 0.75 | cas | 0.45 | 0.44 | 0.64 | 0.71 | cat | 0.47 | 0.54 | 0.71 | 0.77 |
| cav | 0.37 | 0.33 | 0.52 | 0.60 | cax | 0.40 | 0.42 | 0.62 | 0.69 | cbc | 0.42 | 0.39 | 0.58 | 0.65 |
| cbi | 0.44 | 0.45 | 0.63 | 0.70 | cbk | 0.42 | 0.48 | 0.66 | 0.73 | cbr | 0.40 | 0.34 | 0.53 | 0.60 |
| cbs | 0.42 | 0.34 | 0.52 | 0.59 | cbt | 0.46 | 0.31 | 0.49 | 0.57 | cbu | 0.44 | 0.27 | 0.44 | 0.52 |
| cbv | 0.43 | 0.31 | 0.50 | 0.58 | cce | 0.48 | 0.56 | 0.73 | 0.79 | cco | 0.42 | 0.37 | 0.56 | 0.64 |
| ceb | 0.47 | 0.57 | 0.74 | 0.79 | ceg | 0.41 | 0.40 | 0.60 | 0.67 | ces | 0.47 | 0.57 | 0.73 | 0.79 |
| cfm | 0.48 | 0.44 | 0.63 | 0.70 | cgc | 0.47 | 0.46 | 0.65 | 0.72 | cha | 0.46 | 0.58 | 0.75 | 0.80 |
| chd | 0.42 | 0.42 | 0.60 | 0.67 | che | 0.45 | 0.44 | 0.63 | 0.70 | chf | 0.44 | 0.43 | 0.62 | 0.70 |
| chk | 0.46 | 0.50 | 0.69 | 0.76 | chq | 0.43 | 0.38 | 0.57 | 0.65 | chr | 0.46 | 0.51 | 0.69 | 0.75 |
| chu | 0.46 | 0.61 | 0.77 | 0.82 | chv | 0.48 | 0.53 | 0.71 | 0.77 | chz | 0.42 | 0.43 | 0.62 | 0.70 |
| cjo | 0.43 | 0.35 | 0.54 | 0.62 | cjp | 0.45 | 0.50 | 0.68 | 0.75 | cjv | 0.44 | 0.29 | 0.47 | 0.55 |
| ckb | 0.51 | 0.59 | 0.75 | 0.80 | cko | 0.44 | 0.46 | 0.65 | 0.72 | cle | 0.43 | 0.44 | 0.63 | 0.70 |
| clu | 0.44 | 0.51 | 0.69 | 0.75 | cly | 0.42 | 0.33 | 0.52 | 0.61 | cme | 0.41 | 0.42 | 0.61 | 0.68 |
| cmn | 0.49 | 0.60 | 0.76 | 0.82 | cmo | 0.41 | 0.46 | 0.65 | 0.73 | cnh | 0.47 | 0.45 | 0.63 | 0.70 |
| cni | 0.40 | 0.34 | 0.53 | 0.61 | cnl | 0.46 | 0.41 | 0.59 | 0.67 | cnt | 0.41 | 0.44 | 0.62 | 0.69 |
| cnw | 0.45 | 0.45 | 0.63 | 0.70 | coe | 0.43 | 0.34 | 0.53 | 0.61 | cof | 0.42 | 0.38 | 0.57 | 0.65 |
| cok | 0.45 | 0.39 | 0.58 | 0.66 | con | 0.40 | 0.43 | 0.62 | 0.69 | cop | 0.46 | 0.56 | 0.73 | 0.79 |
| cor | 0.50 | 0.60 | 0.76 | 0.81 | cot | 0.44 | 0.43 | 0.61 | 0.68 | cpa | 0.38 | 0.38 | 0.57 | 0.65 |
| cpb | 0.42 | 0.44 | 0.63 | 0.70 | cpc | 0.44 | 0.44 | 0.64 | 0.71 | cpu | 0.44 | 0.45 | 0.64 | 0.71 |
| cpy | 0.45 | 0.43 | 0.62 | 0.70 | crm | 0.44 | 0.54 | 0.71 | 0.77 | crn | 0.47 | 0.39 | 0.58 | 0.66 |
| crq | 0.42 | 0.40 | 0.59 | 0.66 | crs | 0.48 | 0.52 | 0.70 | 0.76 | crt | 0.43 | 0.41 | 0.60 | 0.67 |

Table 10: Transfer performance using other languages as the train/query language (Part I).

| language | classification | retrieval | | | language | classification | retrieval | | | language | classification | retrieval | | |
|---|---|---|---|---|---|---|---|---|---|---|---|---|---|---|
| | | top-1 | top-5 | top-10 | | | top-1 | top-5 | top-10 | | | top-1 | top-5 | top-10 |
| crx | 0.44 | 0.41 | 0.59 | 0.67 | csk | 0.45 | 0.54 | 0.72 | 0.78 | cso | 0.41 | 0.41 | 0.60 | 0.68 |
| csy | 0.47 | 0.47 | 0.65 | 0.72 | cta | 0.43 | 0.29 | 0.48 | 0.56 | ctd | 0.47 | 0.46 | 0.64 | 0.71 |
| ctp | 0.42 | 0.25 | 0.43 | 0.51 | ctu | 0.41 | 0.41 | 0.60 | 0.67 | cub | 0.44 | 0.40 | 0.59 | 0.66 |
| cuc | 0.47 | 0.47 | 0.65 | 0.72 | cui | 0.43 | 0.41 | 0.60 | 0.68 | cuk | 0.43 | 0.47 | 0.66 | 0.72 |
| cul | 0.45 | 0.40 | 0.59 | 0.67 | cut | 0.37 | 0.32 | 0.51 | 0.59 | cux | 0.44 | 0.39 | 0.58 | 0.66 |
| cwe | 0.49 | 0.54 | 0.71 | 0.77 | cwt | 0.45 | 0.54 | 0.72 | 0.77 | cya | 0.40 | 0.34 | 0.54 | 0.62 |
| cym | 0.49 | 0.52 | 0.70 | 0.76 | czt | 0.43 | 0.49 | 0.68 | 0.75 | daa | 0.43 | 0.45 | 0.64 | 0.72 |
| dad | 0.44 | 0.48 | 0.67 | 0.74 | dah | 0.43 | 0.38 | 0.57 | 0.65 | dan | 0.48 | 0.57 | 0.74 | 0.80 |
| ded | 0.47 | 0.47 | 0.66 | 0.73 | des | 0.45 | 0.40 | 0.57 | 0.65 | deu | 0.49 | 0.56 | 0.73 | 0.78 |
| dgc | 0.45 | 0.44 | 0.63 | 0.71 | dgi | 0.47 | 0.49 | 0.67 | 0.74 | dgr | 0.41 | 0.44 | 0.63 | 0.70 |
| dgz | 0.42 | 0.38 | 0.57 | 0.65 | dhm | 0.46 | 0.56 | 0.73 | 0.79 | dig | 0.45 | 0.50 | 0.69 | 0.75 |
| dik | 0.46 | 0.38 | 0.57 | 0.65 | dip | 0.44 | 0.47 | 0.66 | 0.72 | dis | 0.46 | 0.52 | 0.70 | 0.76 |
| dje | 0.47 | 0.56 | 0.74 | 0.80 | djk | 0.42 | 0.34 | 0.53 | 0.61 | djr | 0.44 | 0.42 | 0.61 | 0.68 |
| dnj | 0.45 | 0.35 | 0.53 | 0.61 | dob | 0.45 | 0.47 | 0.66 | 0.73 | dop | 0.43 | 0.42 | 0.61 | 0.68 |
| dow | 0.45 | 0.50 | 0.69 | 0.75 | dtp | 0.44 | 0.51 | 0.70 | 0.76 | dts | 0.46 | 0.52 | 0.70 | 0.77 |
| due | 0.45 | 0.48 | 0.67 | 0.74 | dug | 0.47 | 0.50 | 0.69 | 0.75 | duo | 0.43 | 0.43 | 0.63 | 0.70 |
| dur | 0.45 | 0.41 | 0.61 | 0.69 | dwr | 0.48 | 0.57 | 0.73 | 0.79 | dww | 0.43 | 0.49 | 0.68 | 0.75 |
| dyi | 0.44 | 0.50 | 0.68 | 0.75 | dyo | 0.49 | 0.52 | 0.71 | 0.77 | dyu | 0.46 | 0.49 | 0.68 | 0.74 |
| ebk | 0.47 | 0.54 | 0.72 | 0.78 | efi | 0.48 | 0.48 | 0.65 | 0.72 | eka | 0.48 | 0.48 | 0.67 | 0.74 |
| ell | 0.48 | 0.59 | 0.75 | 0.81 | emp | 0.41 | 0.47 | 0.66 | 0.73 | enb | 0.47 | 0.47 | 0.65 | 0.72 |
| eng | 0.49 | 0.65 | 0.80 | 0.84 | enl | 0.44 | 0.47 | 0.66 | 0.73 | enm | 0.46 | 0.59 | 0.75 | 0.81 |
| epo | 0.47 | 0.61 | 0.77 | 0.82 | eri | 0.42 | 0.42 | 0.61 | 0.68 | ese | 0.39 | 0.29 | 0.47 | 0.55 |
| esi | 0.44 | 0.52 | 0.70 | 0.77 | esk | 0.50 | 0.52 | 0.70 | 0.76 | est | 0.47 | 0.57 | 0.73 | 0.79 |
| esu | 0.46 | 0.54 | 0.72 | 0.78 | etu | 0.43 | 0.45 | 0.64 | 0.72 | eus | 0.50 | 0.59 | 0.75 | 0.81 |
| ewe | 0.50 | 0.50 | 0.68 | 0.74 | eza | 0.46 | 0.44 | 0.63 | 0.70 | faa | 0.44 | 0.36 | 0.54 | 0.61 |
| fai | 0.46 | 0.37 | 0.55 | 0.63 | fal | 0.43 | 0.52 | 0.70 | 0.77 | fao | 0.45 | 0.55 | 0.72 | 0.78 |
| ffm | 0.47 | 0.55 | 0.73 | 0.79 | fij | 0.44 | 0.50 | 0.70 | 0.76 | fil | 0.44 | 0.60 | 0.76 | 0.81 |
| fin | 0.46 | 0.59 | 0.76 | 0.81 | fon | 0.43 | 0.43 | 0.62 | 0.70 | for | 0.42 | 0.43 | 0.62 | 0.69 |
| fra | 0.47 | 0.56 | 0.73 | 0.78 | fry | 0.47 | 0.56 | 0.72 | 0.78 | fub | 0.47 | 0.54 | 0.71 | 0.78 |
| fuf | 0.46 | 0.56 | 0.73 | 0.79 | fuh | 0.50 | 0.55 | 0.73 | 0.79 | fuq | 0.47 | 0.56 | 0.73 | 0.79 |
| fuv | 0.46 | 0.54 | 0.71 | 0.78 | gaa | 0.46 | 0.53 | 0.70 | 0.76 | gag | 0.45 | 0.60 | 0.76 | 0.81 |
| gah | 0.46 | 0.40 | 0.58 | 0.66 | gam | 0.40 | 0.33 | 0.52 | 0.60 | gaw | 0.44 | 0.38 | 0.57 | 0.64 |
| gbi | 0.43 | 0.45 | 0.63 | 0.70 | gbo | 0.47 | 0.43 | 0.62 | 0.69 | gbr | 0.41 | 0.32 | 0.50 | 0.58 |
| gde | 0.43 | 0.50 | 0.68 | 0.75 | gdg | 0.45 | 0.45 | 0.64 | 0.71 | gdn | 0.44 | 0.40 | 0.59 | 0.67 |
| gdr | 0.45 | 0.48 | 0.67 | 0.74 | geb | 0.43 | 0.40 | 0.59 | 0.67 | gej | 0.46 | 0.51 | 0.68 | 0.75 |
| gfk | 0.45 | 0.46 | 0.65 | 0.72 | ghe | 0.49 | 0.54 | 0.72 | 0.78 | ghs | 0.43 | 0.33 | 0.51 | 0.59 |
| gid | 0.44 | 0.46 | 0.65 | 0.73 | gil | 0.48 | 0.51 | 0.69 | 0.75 | giz | 0.47 | 0.45 | 0.64 | 0.71 |
| gjn | 0.42 | 0.52 | 0.70 | 0.76 | gkn | 0.42 | 0.45 | 0.64 | 0.72 | gkp | 0.49 | 0.44 | 0.63 | 0.70 |
| gle | 0.47 | 0.48 | 0.66 | 0.72 | gmv | 0.46 | 0.55 | 0.72 | 0.78 | gnb | 0.46 | 0.49 | 0.67 | 0.74 |
| gnd | 0.37 | 0.37 | 0.57 | 0.65 | gng | 0.41 | 0.52 | 0.70 | 0.77 | gnn | 0.46 | 0.35 | 0.54 | 0.62 |
| gnw | 0.46 | 0.47 | 0.65 | 0.72 | gof | 0.46 | 0.57 | 0.73 | 0.79 | gog | 0.48 | 0.56 | 0.73 | 0.78 |
| gor | 0.47 | 0.52 | 0.69 | 0.75 | gqr | 0.41 | 0.38 | 0.58 | 0.66 | grt | 0.46 | 0.55 | 0.72 | 0.78 |
| gso | 0.47 | 0.48 | 0.67 | 0.73 | gub | 0.37 | 0.29 | 0.47 | 0.55 | guc | 0.40 | 0.38 | 0.56 | 0.64 |
| gud | 0.48 | 0.52 | 0.70 | 0.77 | guh | 0.43 | 0.40 | 0.58 | 0.66 | gui | 0.44 | 0.46 | 0.66 | 0.73 |
| guj | 0.44 | 0.46 | 0.64 | 0.71 | guk | 0.44 | 0.52 | 0.70 | 0.76 | gul | 0.45 | 0.48 | 0.66 | 0.73 |
| gum | 0.45 | 0.48 | 0.66 | 0.72 | gun | 0.41 | 0.49 | 0.67 | 0.74 | guo | 0.40 | 0.37 | 0.56 | 0.63 |
| guq | 0.43 | 0.40 | 0.60 | 0.67 | gur | 0.46 | 0.41 | 0.60 | 0.68 | guw | 0.49 | 0.55 | 0.71 | 0.77 |
| gux | 0.47 | 0.52 | 0.70 | 0.76 | guz | 0.46 | 0.53 | 0.70 | 0.76 | gvc | 0.42 | 0.41 | 0.60 | 0.67 |
| gvf | 0.43 | 0.30 | 0.49 | 0.57 | gvl | 0.49 | 0.44 | 0.63 | 0.71 | gvn | 0.43 | 0.37 | 0.55 | 0.63 |
| gwi | 0.42 | 0.39 | 0.58 | 0.65 | gya | 0.48 | 0.42 | 0.61 | 0.68 | gym | 0.40 | 0.39 | 0.58 | 0.66 |
| gyr | 0.46 | 0.44 | 0.63 | 0.70 | hae | 0.47 | 0.55 | 0.73 | 0.79 | hag | 0.43 | 0.42 | 0.62 | 0.70 |
| hak | 0.44 | 0.58 | 0.75 | 0.81 | hat | 0.47 | 0.41 | 0.61 | 0.68 | hau | 0.49 | 0.56 | 0.72 | 0.78 |
| haw | 0.42 | 0.50 | 0.69 | 0.75 | hay | 0.46 | 0.52 | 0.70 | 0.76 | hch | 0.44 | 0.51 | 0.69 | 0.75 |
| heb | 0.46 | 0.55 | 0.72 | 0.78 | heg | 0.45 | 0.36 | 0.55 | 0.63 | heh | 0.49 | 0.54 | 0.71 | 0.77 |
| hif | 0.46 | 0.45 | 0.63 | 0.70 | hig | 0.43 | 0.47 | 0.67 | 0.74 | hil | 0.47 | 0.60 | 0.77 | 0.82 |
| hin | 0.47 | 0.56 | 0.73 | 0.79 | hix | 0.41 | 0.40 | 0.59 | 0.66 | hla | 0.44 | 0.40 | 0.59 | 0.67 |
| hlt | 0.44 | 0.54 | 0.73 | 0.79 | hmo | 0.46 | 0.53 | 0.71 | 0.77 | hne | 0.48 | 0.55 | 0.72 | 0.78 |
| hnj | 0.50 | 0.45 | 0.64 | 0.72 | hnn | 0.42 | 0.50 | 0.68 | 0.75 | hns | 0.46 | 0.43 | 0.62 | 0.70 |
| hop | 0.45 | 0.49 | 0.67 | 0.74 | hot | 0.45 | 0.37 | 0.56 | 0.64 | hra | 0.44 | 0.49 | 0.67 | 0.74 |
| hrv | 0.46 | 0.58 | 0.75 | 0.80 | hto | 0.42 | 0.43 | 0.62 | 0.69 | hub | 0.44 | 0.37 | 0.56 | 0.64 |
| hui | 0.46 | 0.39 | 0.58 | 0.66 | hun | 0.49 | 0.57 | 0.73 | 0.79 | hus | 0.41 | 0.50 | 0.68 | 0.75 |
| huu | 0.41 | 0.35 | 0.54 | 0.62 | huv | 0.44 | 0.38 | 0.57 | 0.65 | hvn | 0.42 | 0.44 | 0.64 | 0.71 |
| hwc | 0.38 | 0.41 | 0.60 | 0.68 | hye | 0.46 | 0.56 | 0.73 | 0.79 | ian | 0.37 | 0.27 | 0.44 | 0.52 |
| iba | 0.46 | 0.58 | 0.75 | 0.81 | ibo | 0.47 | 0.46 | 0.65 | 0.72 | icr | 0.45 | 0.45 | 0.64 | 0.71 |
| ifa | 0.44 | 0.41 | 0.60 | 0.67 | ifb | 0.44 | 0.43 | 0.62 | 0.69 | ifk | 0.42 | 0.40 | 0.58 | 0.65 |
| ifu | 0.42 | 0.46 | 0.65 | 0.72 | ify | 0.48 | 0.32 | 0.50 | 0.58 | ign | 0.40 | 0.41 | 0.60 | 0.67 |
| ike | 0.47 | 0.55 | 0.72 | 0.78 | ikk | 0.44 | 0.54 | 0.72 | 0.78 | ikw | 0.44 | 0.53 | 0.71 | 0.77 |
| ilb | 0.48 | 0.57 | 0.74 | 0.80 | ilo | 0.47 | 0.56 | 0.73 | 0.78 | imo | 0.47 | 0.34 | 0.53 | 0.61 |
| inb | 0.47 | 0.45 | 0.64 | 0.71 | ind | 0.46 | 0.57 | 0.73 | 0.79 | ino | 0.42 | 0.39 | 0.58 | 0.66 |
| iou | 0.43 | 0.35 | 0.55 | 0.63 | ipi | 0.41 | 0.38 | 0.57 | 0.65 | iqw | 0.45 | 0.45 | 0.64 | 0.71 |
| iri | 0.40 | 0.46 | 0.65 | 0.72 | irk | 0.43 | 0.54 | 0.72 | 0.78 | iry | 0.45 | 0.53 | 0.71 | 0.78 |
| isd | 0.45 | 0.47 | 0.66 | 0.73 | isl | 0.46 | 0.58 | 0.75 | 0.80 | ita | 0.45 | 0.59 | 0.75 | 0.81 |
| itv | 0.46 | 0.56 | 0.74 | 0.79 | ium | 0.44 | 0.42 | 0.61 | 0.69 | ivb | 0.47 | 0.47 | 0.65 | 0.72 |
| ivv | 0.42 | 0.43 | 0.62 | 0.70 | iws | 0.39 | 0.35 | 0.54 | 0.62 | ixl | 0.41 | 0.41 | 0.59 | 0.67 |
| izr | 0.42 | 0.46 | 0.66 | 0.73 | izz | 0.45 | 0.47 | 0.66 | 0.73 | jac | 0.44 | 0.44 | 0.64 | 0.71 |
| jae | 0.45 | 0.49 | 0.68 | 0.75 | jam | 0.43 | 0.41 | 0.60 | 0.68 | jav | 0.47 | 0.48 | 0.67 | 0.73 |
| jbu | 0.45 | 0.37 | 0.56 | 0.64 | jic | 0.36 | 0.28 | 0.46 | 0.54 | jiv | 0.40 | 0.40 | 0.60 | 0.68 |
| jmc | 0.49 | 0.49 | 0.67 | 0.73 | jpn | 0.45 | 0.61 | 0.77 | 0.82 | jra | 0.43 | 0.53 | 0.72 | 0.78 |
| jvn | 0.46 | 0.45 | 0.63 | 0.70 | kaa | 0.45 | 0.54 | 0.71 | 0.77 | kab | 0.45 | 0.51 | 0.69 | 0.75 |
| kac | 0.41 | 0.41 | 0.60 | 0.68 | kal | 0.46 | 0.58 | 0.75 | 0.80 | kan | 0.44 | 0.60 | 0.77 | 0.82 |
| kao | 0.44 | 0.54 | 0.72 | 0.78 | kaq | 0.44 | 0.37 | 0.56 | 0.64 | kat | 0.46 | 0.59 | 0.76 | 0.81 |
| kaz | 0.45 | 0.55 | 0.72 | 0.78 | kbc | 0.40 | 0.47 | 0.66 | 0.73 | kbh | 0.46 | 0.47 | 0.65 | 0.72 |
| kbm | 0.42 | 0.33 | 0.52 | 0.60 | kbp | 0.43 | 0.51 | 0.69 | 0.76 | kbq | 0.43 | 0.44 | 0.62 | 0.70 |
| kbr | 0.43 | 0.53 | 0.71 | 0.76 | kck | 0.52 | 0.56 | 0.73 | 0.79 | kdc | 0.46 | 0.56 | 0.73 | 0.79 |
| kde | 0.48 | 0.59 | 0.75 | 0.81 | kdi | 0.45 | 0.54 | 0.72 | 0.78 | kdj | 0.45 | 0.57 | 0.74 | 0.80 |

Table 11: Transfer performance using other languages as the train/query language (Part II).

| language | classification | retrieval | | | language | classification | retrieval | | | language | classification | retrieval | | |
|---|---|---|---|---|---|---|---|---|---|---|---|---|---|---|
| | | top-1 | top-5 | top-10 | | | top-1 | top-5 | top-10 | | | top-1 | top-5 | top-10 |
| kdl | 0.41 | 0.42 | 0.61 | 0.69 | kek | 0.47 | 0.39 | 0.58 | 0.66 | ken | 0.47 | 0.46 | 0.65 | 0.72 |
| kew | 0.43 | 0.35 | 0.54 | 0.62 | kez | 0.43 | 0.46 | 0.65 | 0.72 | kff | 0.47 | 0.48 | 0.66 | 0.72 |
| kgf | 0.45 | 0.46 | 0.65 | 0.72 | kgk | 0.46 | 0.37 | 0.56 | 0.64 | kgp | 0.41 | 0.30 | 0.49 | 0.58 |
| khk | 0.47 | 0.57 | 0.74 | 0.80 | khm | 0.45 | 0.56 | 0.73 | 0.79 | khs | 0.44 | 0.32 | 0.51 | 0.59 |
| khy | 0.45 | 0.54 | 0.71 | 0.77 | khz | 0.46 | 0.50 | 0.69 | 0.75 | kia | 0.39 | 0.42 | 0.61 | 0.68 |
| kik | 0.48 | 0.54 | 0.71 | 0.77 | kin | 0.46 | 0.55 | 0.72 | 0.78 | kir | 0.46 | 0.53 | 0.70 | 0.76 |
| kix | 0.43 | 0.53 | 0.71 | 0.77 | kjb | 0.40 | 0.39 | 0.59 | 0.66 | kje | 0.42 | 0.38 | 0.58 | 0.66 |
| kjh | 0.46 | 0.52 | 0.70 | 0.76 | kjs | 0.43 | 0.37 | 0.56 | 0.64 | kki | 0.45 | 0.54 | 0.71 | 0.77 |
| kkj | 0.45 | 0.46 | 0.65 | 0.72 | klv | 0.43 | 0.49 | 0.67 | 0.74 | kma | 0.44 | 0.48 | 0.67 | 0.73 |
| kmg | 0.42 | 0.44 | 0.63 | 0.70 | kmh | 0.40 | 0.27 | 0.45 | 0.53 | kmk | 0.45 | 0.50 | 0.68 | 0.75 |
| kmm | 0.47 | 0.48 | 0.67 | 0.73 | kmo | 0.43 | 0.35 | 0.54 | 0.62 | kmr | 0.47 | 0.59 | 0.75 | 0.80 |
| kms | 0.41 | 0.32 | 0.51 | 0.59 | kmu | 0.40 | 0.39 | 0.59 | 0.66 | kne | 0.44 | 0.50 | 0.68 | 0.75 |
| knf | 0.44 | 0.52 | 0.71 | 0.77 | kng | 0.45 | 0.52 | 0.70 | 0.76 | knj | 0.40 | 0.44 | 0.63 | 0.71 |
| knk | 0.45 | 0.50 | 0.68 | 0.75 | kno | 0.37 | 0.34 | 0.53 | 0.61 | knv | 0.42 | 0.39 | 0.58 | 0.65 |
| kog | 0.40 | 0.41 | 0.60 | 0.67 | kor | 0.46 | 0.57 | 0.74 | 0.79 | kpf | 0.43 | 0.39 | 0.58 | 0.65 |
| kpg | 0.41 | 0.39 | 0.58 | 0.66 | kpj | 0.45 | 0.36 | 0.54 | 0.62 | kpr | 0.45 | 0.35 | 0.53 | 0.61 |
| kpv | 0.46 | 0.55 | 0.73 | 0.78 | kpw | 0.41 | 0.27 | 0.45 | 0.53 | kpx | 0.42 | 0.35 | 0.54 | 0.62 |
| kpz | 0.44 | 0.53 | 0.71 | 0.77 | kqe | 0.48 | 0.52 | 0.70 | 0.77 | kqo | 0.44 | 0.45 | 0.64 | 0.71 |
| kqp | 0.41 | 0.38 | 0.57 | 0.65 | kqs | 0.43 | 0.53 | 0.71 | 0.77 | kqy | 0.42 | 0.53 | 0.70 | 0.77 |
| krc | 0.47 | 0.57 | 0.74 | 0.79 | kri | 0.42 | 0.43 | 0.62 | 0.70 | krj | 0.43 | 0.57 | 0.74 | 0.80 |
| ksc | 0.45 | 0.45 | 0.64 | 0.71 | ksd | 0.46 | 0.50 | 0.68 | 0.75 | ksf | 0.48 | 0.51 | 0.69 | 0.76 |
| ksr | 0.43 | 0.44 | 0.63 | 0.70 | kss | 0.43 | 0.45 | 0.64 | 0.71 | ksw | 0.48 | 0.51 | 0.69 | 0.75 |
| ktb | 0.45 | 0.56 | 0.73 | 0.79 | ktj | 0.43 | 0.40 | 0.60 | 0.68 | kto | 0.42 | 0.41 | 0.61 | 0.68 |
| ktu | 0.47 | 0.55 | 0.72 | 0.78 | kua | 0.46 | 0.54 | 0.71 | 0.77 | kub | 0.44 | 0.40 | 0.60 | 0.67 |
| kud | 0.46 | 0.45 | 0.63 | 0.70 | kue | 0.47 | 0.42 | 0.62 | 0.69 | kum | 0.42 | 0.52 | 0.70 | 0.76 |
| kup | 0.41 | 0.31 | 0.49 | 0.57 | kus | 0.45 | 0.52 | 0.70 | 0.77 | kvj | 0.44 | 0.50 | 0.69 | 0.76 |
| kvn | 0.44 | 0.42 | 0.60 | 0.68 | kwd | 0.42 | 0.47 | 0.66 | 0.73 | kwf | 0.41 | 0.48 | 0.66 | 0.73 |
| kwi | 0.41 | 0.37 | 0.56 | 0.64 | kwj | 0.42 | 0.36 | 0.55 | 0.63 | kxc | 0.49 | 0.53 | 0.71 | 0.77 |
| kxm | 0.46 | 0.46 | 0.66 | 0.73 | kxw | 0.43 | 0.37 | 0.57 | 0.65 | kyc | 0.43 | 0.36 | 0.55 | 0.63 |
| kyf | 0.45 | 0.49 | 0.67 | 0.73 | kyg | 0.42 | 0.43 | 0.61 | 0.69 | kyq | 0.43 | 0.43 | 0.63 | 0.70 |
| kyu | 0.41 | 0.53 | 0.71 | 0.77 | kyz | 0.40 | 0.33 | 0.52 | 0.60 | kze | 0.47 | 0.39 | 0.58 | 0.65 |
| lac | 0.38 | 0.34 | 0.53 | 0.62 | lai | 0.46 | 0.57 | 0.75 | 0.80 | laj | 0.46 | 0.53 | 0.71 | 0.77 |
| lam | 0.43 | 0.56 | 0.73 | 0.78 | lao | 0.47 | 0.56 | 0.73 | 0.79 | las | 0.43 | 0.45 | 0.64 | 0.71 |
| lat | 0.46 | 0.57 | 0.74 | 0.80 | lav | 0.45 | 0.58 | 0.75 | 0.80 | lbk | 0.46 | 0.53 | 0.71 | 0.77 |
| lcm | 0.47 | 0.39 | 0.58 | 0.66 | ldi | 0.47 | 0.49 | 0.68 | 0.74 | lee | 0.48 | 0.44 | 0.63 | 0.70 |
| lef | 0.44 | 0.40 | 0.60 | 0.67 | leh | 0.46 | 0.51 | 0.69 | 0.75 | lem | 0.45 | 0.46 | 0.64 | 0.71 |
| leu | 0.47 | 0.45 | 0.65 | 0.72 | lew | 0.46 | 0.50 | 0.69 | 0.76 | lex | 0.46 | 0.41 | 0.60 | 0.68 |
| lgm | 0.43 | 0.55 | 0.72 | 0.78 | lhi | 0.44 | 0.39 | 0.59 | 0.67 | lhm | 0.43 | 0.49 | 0.68 | 0.74 |
| lhu | 0.48 | 0.42 | 0.62 | 0.70 | lia | 0.47 | 0.51 | 0.69 | 0.76 | lid | 0.43 | 0.35 | 0.54 | 0.62 |
| lif | 0.49 | 0.48 | 0.67 | 0.73 | lin | 0.47 | 0.53 | 0.71 | 0.77 | lip | 0.39 | 0.42 | 0.60 | 0.67 |
| lit | 0.49 | 0.58 | 0.75 | 0.81 | ljp | 0.48 | 0.54 | 0.71 | 0.78 | lmk | 0.45 | 0.52 | 0.69 | 0.76 |
| lmp | 0.46 | 0.47 | 0.66 | 0.74 | lob | 0.47 | 0.47 | 0.66 | 0.72 | lol | 0.48 | 0.50 | 0.68 | 0.74 |
| lom | 0.46 | 0.44 | 0.63 | 0.71 | loz | 0.48 | 0.52 | 0.70 | 0.76 | lsi | 0.42 | 0.40 | 0.59 | 0.67 |
| lsm | 0.47 | 0.52 | 0.70 | 0.76 | lug | 0.45 | 0.57 | 0.74 | 0.79 | luo | 0.46 | 0.50 | 0.68 | 0.74 |
| lus | 0.41 | 0.54 | 0.71 | 0.77 | lwo | 0.41 | 0.37 | 0.56 | 0.64 | lww | 0.45 | 0.31 | 0.50 | 0.58 |
| lzh | 0.47 | 0.62 | 0.77 | 0.82 | maa | 0.44 | 0.47 | 0.66 | 0.73 | mad | 0.46 | 0.52 | 0.70 | 0.76 |
| maf | 0.47 | 0.50 | 0.69 | 0.75 | mah | 0.46 | 0.50 | 0.69 | 0.75 | mai | 0.44 | 0.57 | 0.74 | 0.80 |
| maj | 0.45 | 0.50 | 0.69 | 0.75 | mak | 0.44 | 0.52 | 0.70 | 0.77 | mal | 0.45 | 0.56 | 0.72 | 0.78 |
| mam | 0.44 | 0.48 | 0.67 | 0.74 | maq | 0.40 | 0.41 | 0.59 | 0.67 | mar | 0.44 | 0.50 | 0.68 | 0.75 |
| mau | 0.45 | 0.40 | 0.59 | 0.67 | mav | 0.40 | 0.28 | 0.46 | 0.55 | maw | 0.45 | 0.47 | 0.66 | 0.73 |
| maz | 0.43 | 0.38 | 0.58 | 0.65 | mbb | 0.44 | 0.48 | 0.67 | 0.74 | mbc | 0.45 | 0.37 | 0.55 | 0.62 |
| mbd | 0.45 | 0.42 | 0.61 | 0.69 | mbf | 0.47 | 0.59 | 0.75 | 0.80 | mbh | 0.46 | 0.45 | 0.64 | 0.71 |
| mbi | 0.43 | 0.45 | 0.64 | 0.71 | mbj | 0.41 | 0.41 | 0.61 | 0.68 | mbl | 0.45 | 0.29 | 0.48 | 0.57 |
| mbs | 0.43 | 0.45 | 0.64 | 0.71 | mbt | 0.45 | 0.53 | 0.71 | 0.77 | mca | 0.45 | 0.48 | 0.67 | 0.74 |
| mcb | 0.39 | 0.39 | 0.58 | 0.65 | mcd | 0.42 | 0.33 | 0.51 | 0.59 | mcf | 0.41 | 0.22 | 0.39 | 0.47 |
| mck | 0.47 | 0.51 | 0.69 | 0.76 | mcn | 0.45 | 0.50 | 0.69 | 0.75 | mco | 0.40 | 0.42 | 0.60 | 0.68 |
| mcp | 0.45 | 0.50 | 0.68 | 0.75 | mcq | 0.45 | 0.42 | 0.62 | 0.69 | mcu | 0.44 | 0.39 | 0.58 | 0.66 |
| mda | 0.43 | 0.34 | 0.53 | 0.61 | mdy | 0.46 | 0.51 | 0.69 | 0.75 | med | 0.44 | 0.28 | 0.46 | 0.54 |
| mee | 0.43 | 0.48 | 0.67 | 0.73 | mej | 0.42 | 0.34 | 0.54 | 0.62 | mek | 0.42 | 0.43 | 0.63 | 0.70 |
| men | 0.45 | 0.48 | 0.67 | 0.73 | meq | 0.43 | 0.40 | 0.61 | 0.68 | meu | 0.46 | 0.48 | 0.67 | 0.74 |
| mfe | 0.47 | 0.54 | 0.71 | 0.77 | mfh | 0.46 | 0.40 | 0.59 | 0.66 | mfi | 0.45 | 0.43 | 0.62 | 0.69 |
| mfk | 0.44 | 0.44 | 0.63 | 0.70 | mfq | 0.45 | 0.48 | 0.66 | 0.73 | mfy | 0.45 | 0.44 | 0.62 | 0.69 |
| mfz | 0.43 | 0.43 | 0.62 | 0.69 | mhi | 0.44 | 0.40 | 0.59 | 0.66 | mhl | 0.44 | 0.38 | 0.57 | 0.64 |
| mhr | 0.45 | 0.57 | 0.74 | 0.79 | mhx | 0.45 | 0.39 | 0.59 | 0.66 | mhy | 0.43 | 0.52 | 0.70 | 0.76 |
| mib | 0.41 | 0.44 | 0.63 | 0.70 | mic | 0.46 | 0.44 | 0.63 | 0.70 | mie | 0.48 | 0.45 | 0.64 | 0.71 |
| mif | 0.44 | 0.45 | 0.64 | 0.71 | mig | 0.44 | 0.49 | 0.67 | 0.74 | mih | 0.43 | 0.35 | 0.55 | 0.63 |
| mil | 0.40 | 0.33 | 0.51 | 0.59 | min | 0.45 | 0.52 | 0.70 | 0.76 | mio | 0.39 | 0.32 | 0.52 | 0.60 |
| miq | 0.45 | 0.42 | 0.62 | 0.69 | mir | 0.38 | 0.34 | 0.51 | 0.59 | mit | 0.43 | 0.40 | 0.58 | 0.66 |
| miy | 0.42 | 0.42 | 0.62 | 0.70 | miz | 0.40 | 0.35 | 0.55 | 0.63 | mjc | 0.42 | 0.36 | 0.55 | 0.63 |
| mjw | 0.49 | 0.53 | 0.71 | 0.78 | mkd | 0.46 | 0.59 | 0.75 | 0.81 | mkl | 0.44 | 0.48 | 0.67 | 0.73 |
| mkn | 0.43 | 0.38 | 0.57 | 0.65 | mks | 0.44 | 0.38 | 0.58 | 0.66 | mlh | 0.48 | 0.43 | 0.63 | 0.70 |
| mlp | 0.45 | 0.40 | 0.59 | 0.67 | mlt | 0.49 | 0.57 | 0.74 | 0.79 | mmn | 0.44 | 0.43 | 0.62 | 0.70 |
| mmo | 0.46 | 0.41 | 0.59 | 0.67 | mmx | 0.41 | 0.44 | 0.63 | 0.70 | mna | 0.40 | 0.36 | 0.56 | 0.64 |
| mnb | 0.44 | 0.56 | 0.74 | 0.79 | mnf | 0.45 | 0.42 | 0.61 | 0.69 | mnh | 0.44 | 0.44 | 0.63 | 0.70 |
| mnk | 0.46 | 0.57 | 0.74 | 0.80 | mnx | 0.41 | 0.35 | 0.54 | 0.62 | moc | 0.44 | 0.48 | 0.65 | 0.72 |
| mog | 0.41 | 0.48 | 0.66 | 0.72 | mop | 0.41 | 0.40 | 0.59 | 0.67 | mor | 0.46 | 0.50 | 0.69 | 0.75 |
| mos | 0.46 | 0.44 | 0.63 | 0.70 | mox | 0.46 | 0.47 | 0.66 | 0.72 | mpg | 0.45 | 0.49 | 0.68 | 0.74 |
| mpm | 0.44 | 0.31 | 0.50 | 0.58 | mps | 0.41 | 0.26 | 0.43 | 0.52 | mpt | 0.41 | 0.35 | 0.53 | 0.61 |
| mqb | 0.47 | 0.38 | 0.58 | 0.66 | mqj | 0.45 | 0.53 | 0.71 | 0.77 | mqy | 0.46 | 0.49 | 0.68 | 0.74 |
| mri | 0.45 | 0.45 | 0.64 | 0.71 | mrw | 0.47 | 0.55 | 0.72 | 0.78 | msa | 0.47 | 0.58 | 0.75 | 0.81 |
| msb | 0.45 | 0.55 | 0.72 | 0.78 | mse | 0.48 | 0.44 | 0.63 | 0.70 | msk | 0.45 | 0.41 | 0.61 | 0.68 |
| msm | 0.43 | 0.49 | 0.67 | 0.74 | msy | 0.44 | 0.44 | 0.63 | 0.70 | mta | 0.44 | 0.44 | 0.63 | 0.70 |
| mtg | 0.44 | 0.36 | 0.55 | 0.63 | mti | 0.44 | 0.46 | 0.65 | 0.72 | mtj | 0.43 | 0.34 | 0.53 | 0.62 |
| mto | 0.40 | 0.47 | 0.66 | 0.73 | mtp | 0.47 | 0.50 | 0.68 | 0.74 | mua | 0.38 | 0.46 | 0.65 | 0.72 |
| muh | 0.38 | 0.22 | 0.39 | 0.48 | mur | 0.44 | 0.46 | 0.65 | 0.72 | mux | 0.42 | 0.37 | 0.55 | 0.63 |

Table 12: Transfer performance using other languages as the train/query language (Part III).

| language | classification | retrieval | | | language | classification | retrieval | | | language | classification | retrieval | | |
|---|---|---|---|---|---|---|---|---|---|---|---|---|---|---|
| | | top-1 | top-5 | top-10 | | | top-1 | top-5 | top-10 | | | top-1 | top-5 | top-10 |
| muy | 0.43 | 0.43 | 0.62 | 0.69 | mva | 0.45 | 0.47 | 0.66 | 0.72 | mvn | 0.45 | 0.44 | 0.63 | 0.70 |
| mvp | 0.45 | 0.49 | 0.68 | 0.74 | mwm | 0.44 | 0.40 | 0.59 | 0.67 | mwq | 0.45 | 0.45 | 0.65 | 0.72 |
| mwv | 0.45 | 0.52 | 0.70 | 0.76 | mww | 0.40 | 0.45 | 0.65 | 0.72 | mxb | 0.39 | 0.42 | 0.61 | 0.69 |
| mxp | 0.44 | 0.37 | 0.56 | 0.63 | mxq | 0.43 | 0.45 | 0.64 | 0.71 | mxt | 0.43 | 0.37 | 0.56 | 0.64 |
| mya | 0.48 | 0.60 | 0.76 | 0.81 | myb | 0.43 | 0.48 | 0.67 | 0.73 | myk | 0.44 | 0.45 | 0.63 | 0.70 |
| myu | 0.41 | 0.38 | 0.58 | 0.65 | myv | 0.50 | 0.54 | 0.71 | 0.77 | myw | 0.40 | 0.41 | 0.60 | 0.68 |
| myx | 0.49 | 0.54 | 0.71 | 0.77 | myy | 0.44 | 0.33 | 0.51 | 0.59 | mza | 0.42 | 0.38 | 0.57 | 0.65 |
| mzh | 0.46 | 0.44 | 0.63 | 0.70 | mzk | 0.44 | 0.39 | 0.58 | 0.66 | mzl | 0.43 | 0.45 | 0.64 | 0.71 |
| mzm | 0.40 | 0.35 | 0.55 | 0.63 | mzw | 0.45 | 0.43 | 0.62 | 0.70 | nab | 0.38 | 0.34 | 0.53 | 0.61 |
| naf | 0.44 | 0.34 | 0.52 | 0.60 | nak | 0.42 | 0.32 | 0.51 | 0.59 | nan | 0.48 | 0.57 | 0.74 | 0.80 |
| naq | 0.49 | 0.57 | 0.74 | 0.80 | nas | 0.46 | 0.46 | 0.64 | 0.71 | nav | 0.46 | 0.50 | 0.69 | 0.76 |
| nbc | 0.45 | 0.54 | 0.72 | 0.78 | nbe | 0.46 | 0.55 | 0.72 | 0.78 | nbl | 0.50 | 0.57 | 0.73 | 0.79 |
| nca | 0.44 | 0.43 | 0.63 | 0.70 | nch | 0.45 | 0.47 | 0.65 | 0.72 | ncj | 0.42 | 0.45 | 0.63 | 0.70 |
| ncl | 0.44 | 0.40 | 0.58 | 0.66 | nct | 0.47 | 0.44 | 0.63 | 0.70 | ncu | 0.45 | 0.41 | 0.60 | 0.67 |
| ndc | 0.48 | 0.59 | 0.76 | 0.81 | nde | 0.49 | 0.56 | 0.73 | 0.79 | ndi | 0.43 | 0.46 | 0.65 | 0.72 |
| ndj | 0.45 | 0.55 | 0.72 | 0.78 | ndo | 0.46 | 0.55 | 0.72 | 0.78 | ndp | 0.51 | 0.52 | 0.70 | 0.76 |
| nds | 0.47 | 0.54 | 0.71 | 0.77 | ndz | 0.41 | 0.34 | 0.53 | 0.61 | neb | 0.42 | 0.43 | 0.62 | 0.69 |
| nep | 0.48 | 0.61 | 0.78 | 0.83 | nfa | 0.42 | 0.37 | 0.57 | 0.65 | nfr | 0.43 | 0.43 | 0.62 | 0.69 |
| ngc | 0.46 | 0.55 | 0.72 | 0.78 | ngp | 0.49 | 0.51 | 0.69 | 0.76 | ngu | 0.44 | 0.49 | 0.67 | 0.74 |
| nhd | 0.43 | 0.49 | 0.67 | 0.74 | nhe | 0.44 | 0.47 | 0.66 | 0.73 | nhg | 0.42 | 0.52 | 0.70 | 0.76 |
| nhi | 0.45 | 0.53 | 0.70 | 0.77 | nho | 0.47 | 0.39 | 0.59 | 0.66 | nhr | 0.47 | 0.54 | 0.72 | 0.78 |
| nhu | 0.45 | 0.46 | 0.65 | 0.72 | nhw | 0.42 | 0.47 | 0.66 | 0.72 | nhx | 0.41 | 0.47 | 0.66 | 0.73 |
| nhy | 0.46 | 0.51 | 0.69 | 0.75 | nii | 0.40 | 0.24 | 0.41 | 0.49 | nij | 0.48 | 0.50 | 0.68 | 0.75 |
| nim | 0.46 | 0.58 | 0.74 | 0.79 | nin | 0.44 | 0.36 | 0.55 | 0.63 | niq | 0.46 | 0.60 | 0.76 | 0.82 |
| niy | 0.43 | 0.51 | 0.69 | 0.75 | njb | 0.43 | 0.50 | 0.68 | 0.75 | njm | 0.43 | 0.50 | 0.67 | 0.74 |
| njn | 0.41 | 0.53 | 0.70 | 0.76 | njo | 0.45 | 0.56 | 0.73 | 0.78 | njz | 0.46 | 0.51 | 0.69 | 0.76 |
| nko | 0.45 | 0.50 | 0.69 | 0.75 | nlc | 0.43 | 0.38 | 0.57 | 0.64 | nld | 0.47 | 0.55 | 0.72 | 0.78 |
| nma | 0.44 | 0.53 | 0.70 | 0.76 | nmf | 0.44 | 0.51 | 0.69 | 0.75 | nmo | 0.48 | 0.50 | 0.68 | 0.75 |
| nmz | 0.40 | 0.48 | 0.67 | 0.73 | nnb | 0.46 | 0.59 | 0.76 | 0.81 | nng | 0.45 | 0.52 | 0.70 | 0.76 |
| nnh | 0.42 | 0.38 | 0.58 | 0.66 | nno | 0.46 | 0.55 | 0.72 | 0.78 | nnp | 0.48 | 0.48 | 0.66 | 0.73 |
| nnq | 0.46 | 0.53 | 0.71 | 0.77 | nnw | 0.42 | 0.43 | 0.63 | 0.70 | noa | 0.42 | 0.34 | 0.52 | 0.60 |
| nob | 0.48 | 0.58 | 0.74 | 0.80 | nog | 0.49 | 0.53 | 0.70 | 0.76 | nop | 0.42 | 0.41 | 0.60 | 0.68 |
| not | 0.42 | 0.37 | 0.56 | 0.64 | nou | 0.43 | 0.34 | 0.53 | 0.61 | nph | 0.46 | 0.56 | 0.74 | 0.79 |
| npi | 0.45 | 0.58 | 0.74 | 0.80 | npl | 0.46 | 0.49 | 0.67 | 0.73 | npo | 0.46 | 0.50 | 0.69 | 0.76 |
| npy | 0.44 | 0.49 | 0.67 | 0.74 | nsn | 0.46 | 0.46 | 0.65 | 0.72 | nso | 0.48 | 0.54 | 0.71 | 0.77 |
| ntp | 0.42 | 0.38 | 0.56 | 0.64 | ntr | 0.44 | 0.40 | 0.60 | 0.67 | nus | 0.47 | 0.50 | 0.68 | 0.74 |
| nuy | 0.48 | 0.44 | 0.63 | 0.70 | nvm | 0.45 | 0.34 | 0.52 | 0.59 | nwb | 0.42 | 0.39 | 0.58 | 0.65 |
| nwi | 0.44 | 0.39 | 0.58 | 0.66 | nya | 0.47 | 0.56 | 0.73 | 0.79 | nyf | 0.43 | 0.54 | 0.72 | 0.78 |
| nyn | 0.47 | 0.55 | 0.73 | 0.79 | nyo | 0.44 | 0.55 | 0.73 | 0.78 | nyy | 0.49 | 0.56 | 0.73 | 0.79 |
| obo | 0.47 | 0.50 | 0.68 | 0.75 | oji | 0.46 | 0.45 | 0.64 | 0.71 | ojs | 0.48 | 0.53 | 0.72 | 0.78 |
| okv | 0.43 | 0.32 | 0.51 | 0.59 | old | 0.45 | 0.50 | 0.68 | 0.74 | omw | 0.43 | 0.35 | 0.54 | 0.62 |
| ong | 0.42 | 0.31 | 0.50 | 0.58 | ons | 0.41 | 0.39 | 0.59 | 0.67 | ood | 0.44 | 0.36 | 0.55 | 0.63 |
| opm | 0.42 | 0.33 | 0.51 | 0.59 | ory | 0.44 | 0.52 | 0.69 | 0.76 | oss | 0.47 | 0.51 | 0.69 | 0.75 |
| ote | 0.47 | 0.43 | 0.62 | 0.69 | otm | 0.44 | 0.37 | 0.56 | 0.64 | otn | 0.40 | 0.42 | 0.61 | 0.68 |
| otq | 0.45 | 0.45 | 0.64 | 0.71 | ots | 0.43 | 0.31 | 0.49 | 0.57 | ozm | 0.47 | 0.46 | 0.65 | 0.72 |
| pab | 0.46 | 0.43 | 0.62 | 0.69 | pad | 0.42 | 0.41 | 0.61 | 0.68 | pag | 0.50 | 0.61 | 0.77 | 0.82 |
| pah | 0.41 | 0.39 | 0.59 | 0.66 | pam | 0.46 | 0.56 | 0.73 | 0.79 | pan | 0.47 | 0.54 | 0.72 | 0.78 |
| pao | 0.43 | 0.23 | 0.41 | 0.49 | pap | 0.47 | 0.58 | 0.75 | 0.80 | pbb | 0.42 | 0.39 | 0.58 | 0.65 |
| pbc | 0.46 | 0.47 | 0.66 | 0.73 | pbi | 0.41 | 0.44 | 0.63 | 0.70 | pbl | 0.48 | 0.52 | 0.70 | 0.76 |
| pcm | 0.48 | 0.48 | 0.67 | 0.73 | pdc | 0.47 | 0.53 | 0.71 | 0.77 | pdt | 0.48 | 0.52 | 0.70 | 0.77 |
| pes | 0.45 | 0.57 | 0.73 | 0.79 | pib | 0.45 | 0.57 | 0.75 | 0.80 | pio | 0.41 | 0.40 | 0.59 | 0.67 |
| pir | 0.46 | 0.39 | 0.58 | 0.65 | pis | 0.44 | 0.52 | 0.71 | 0.77 | pkb | 0.47 | 0.53 | 0.71 | 0.77 |
| plg | 0.42 | 0.47 | 0.66 | 0.73 | pls | 0.41 | 0.42 | 0.60 | 0.68 | plu | 0.44 | 0.42 | 0.61 | 0.68 |
| plw | 0.43 | 0.48 | 0.66 | 0.73 | pmf | 0.45 | 0.47 | 0.66 | 0.72 | pne | 0.45 | 0.45 | 0.65 | 0.72 |
| poe | 0.41 | 0.46 | 0.65 | 0.72 | poh | 0.47 | 0.39 | 0.59 | 0.66 | poi | 0.43 | 0.48 | 0.66 | 0.73 |
| pol | 0.47 | 0.57 | 0.73 | 0.79 | pon | 0.49 | 0.49 | 0.68 | 0.74 | por | 0.51 | 0.59 | 0.75 | 0.81 |
| poy | 0.45 | 0.56 | 0.73 | 0.79 | ppk | 0.48 | 0.50 | 0.69 | 0.75 | ppo | 0.46 | 0.38 | 0.57 | 0.65 |
| prf | 0.47 | 0.56 | 0.74 | 0.80 | pri | 0.46 | 0.45 | 0.65 | 0.72 | prk | 0.47 | 0.49 | 0.67 | 0.74 |
| prs | 0.44 | 0.55 | 0.73 | 0.78 | pse | 0.42 | 0.55 | 0.73 | 0.79 | ptp | 0.41 | 0.35 | 0.55 | 0.63 |
| ptu | 0.45 | 0.53 | 0.71 | 0.77 | pua | 0.45 | 0.45 | 0.63 | 0.71 | pwg | 0.44 | 0.46 | 0.66 | 0.73 |
| pww | 0.41 | 0.46 | 0.65 | 0.72 | qub | 0.43 | 0.37 | 0.56 | 0.64 | quc | 0.45 | 0.40 | 0.59 | 0.67 |
| quf | 0.40 | 0.41 | 0.59 | 0.66 | qug | 0.44 | 0.51 | 0.69 | 0.75 | quh | 0.47 | 0.53 | 0.70 | 0.77 |
| qul | 0.45 | 0.56 | 0.73 | 0.79 | qup | 0.48 | 0.38 | 0.57 | 0.65 | quw | 0.43 | 0.55 | 0.73 | 0.79 |
| quy | 0.45 | 0.50 | 0.69 | 0.75 | quz | 0.45 | 0.55 | 0.73 | 0.79 | qvc | 0.41 | 0.42 | 0.61 | 0.68 |
| qve | 0.45 | 0.49 | 0.67 | 0.74 | qvh | 0.41 | 0.34 | 0.52 | 0.60 | qvi | 0.48 | 0.51 | 0.70 | 0.76 |
| qvm | 0.40 | 0.34 | 0.52 | 0.60 | qvn | 0.43 | 0.41 | 0.59 | 0.66 | qvo | 0.44 | 0.46 | 0.64 | 0.70 |
| qvs | 0.40 | 0.41 | 0.59 | 0.66 | qvw | 0.42 | 0.45 | 0.64 | 0.71 | qvz | 0.43 | 0.42 | 0.60 | 0.68 |
| qwh | 0.47 | 0.48 | 0.66 | 0.72 | qxh | 0.40 | 0.36 | 0.55 | 0.62 | qxn | 0.44 | 0.47 | 0.67 | 0.73 |
| qxo | 0.42 | 0.32 | 0.50 | 0.58 | qxr | 0.42 | 0.50 | 0.69 | 0.76 | rai | 0.43 | 0.46 | 0.66 | 0.73 |
| rim | 0.47 | 0.51 | 0.69 | 0.76 | rkb | 0.40 | 0.30 | 0.48 | 0.56 | rmo | 0.42 | 0.40 | 0.59 | 0.67 |
| rmy | 0.46 | 0.52 | 0.69 | 0.75 | ron | 0.47 | 0.56 | 0.72 | 0.78 | roo | 0.46 | 0.42 | 0.61 | 0.68 |
| rop | 0.42 | 0.35 | 0.54 | 0.62 | rro | 0.43 | 0.49 | 0.67 | 0.74 | ruf | 0.44 | 0.55 | 0.73 | 0.79 |
| run | 0.50 | 0.54 | 0.72 | 0.78 | rus | 0.48 | 0.55 | 0.72 | 0.78 | rwo | 0.46 | 0.39 | 0.57 | 0.65 |
| sab | 0.37 | 0.28 | 0.47 | 0.55 | sag | 0.46 | 0.48 | 0.67 | 0.73 | sah | 0.46 | 0.53 | 0.71 | 0.77 |
| sas | 0.50 | 0.55 | 0.72 | 0.78 | sba | 0.45 | 0.40 | 0.59 | 0.66 | sbd | 0.44 | 0.41 | 0.60 | 0.68 |
| sbl | 0.43 | 0.51 | 0.69 | 0.76 | sda | 0.46 | 0.56 | 0.73 | 0.79 | sey | 0.43 | 0.43 | 0.63 | 0.70 |
| sgb | 0.42 | 0.43 | 0.61 | 0.69 | sgw | 0.45 | 0.51 | 0.70 | 0.76 | sgz | 0.41 | 0.38 | 0.57 | 0.65 |
| shi | 0.45 | 0.51 | 0.69 | 0.75 | shk | 0.44 | 0.48 | 0.67 | 0.73 | shp | 0.46 | 0.44 | 0.62 | 0.69 |
| shu | 0.45 | 0.50 | 0.69 | 0.75 | sig | 0.41 | 0.44 | 0.63 | 0.71 | sil | 0.43 | 0.44 | 0.64 | 0.71 |
| sim | 0.44 | 0.37 | 0.56 | 0.64 | sin | 0.48 | 0.49 | 0.66 | 0.73 | sja | 0.43 | 0.37 | 0.55 | 0.63 |
| sld | 0.46 | 0.47 | 0.66 | 0.73 | slk | 0.48 | 0.58 | 0.74 | 0.80 | sll | 0.39 | 0.24 | 0.42 | 0.51 |
| slv | 0.49 | 0.56 | 0.73 | 0.79 | sme | 0.47 | 0.62 | 0.78 | 0.83 | smk | 0.39 | 0.47 | 0.65 | 0.72 |
| sml | 0.45 | 0.50 | 0.69 | 0.76 | smo | 0.46 | 0.52 | 0.70 | 0.76 | sna | 0.49 | 0.56 | 0.73 | 0.79 |
| snc | 0.47 | 0.52 | 0.70 | 0.76 | snd | 0.44 | 0.55 | 0.73 | 0.78 | snn | 0.42 | 0.30 | 0.49 | 0.57 |
| snp | 0.43 | 0.42 | 0.61 | 0.68 | snw | 0.45 | 0.46 | 0.64 | 0.71 | sny | 0.38 | 0.30 | 0.49 | 0.57 |

Table 13: Transfer performance using other languages as the train/query language (Part IV).

| language | classification | retrieval | | | language | classification | retrieval | | | language | classification | retrieval | | |
|---|---|---|---|---|---|---|---|---|---|---|---|---|---|---|
| | | top-1 | top-5 | top-10 | | | top-1 | top-5 | top-10 | | | top-1 | top-5 | top-10 |
| som | 0.48 | 0.57 | 0.74 | 0.80 | soq | 0.47 | 0.43 | 0.62 | 0.69 | sot | 0.47 | 0.52 | 0.69 | 0.75 |
| soy | 0.45 | 0.48 | 0.67 | 0.74 | spa | 0.47 | 0.58 | 0.74 | 0.80 | spl | 0.38 | 0.25 | 0.42 | 0.50 |
| spp | 0.47 | 0.48 | 0.66 | 0.73 | sps | 0.44 | 0.42 | 0.60 | 0.67 | spy | 0.47 | 0.48 | 0.67 | 0.73 |
| sqi | 0.42 | 0.32 | 0.45 | 0.51 | sri | 0.42 | 0.43 | 0.62 | 0.69 | srm | 0.43 | 0.34 | 0.53 | 0.61 |
| srn | 0.48 | 0.50 | 0.68 | 0.74 | srp | 0.46 | 0.59 | 0.75 | 0.80 | srq | 0.41 | 0.31 | 0.50 | 0.59 |
| ssd | 0.47 | 0.38 | 0.58 | 0.65 | ssg | 0.41 | 0.37 | 0.56 | 0.64 | ssw | 0.47 | 0.57 | 0.74 | 0.80 |
| ssx | 0.45 | 0.39 | 0.58 | 0.65 | stn | 0.45 | 0.43 | 0.62 | 0.69 | stp | 0.39 | 0.27 | 0.45 | 0.53 |
| sua | 0.43 | 0.38 | 0.56 | 0.64 | sue | 0.42 | 0.36 | 0.56 | 0.64 | suk | 0.43 | 0.54 | 0.71 | 0.77 |
| sun | 0.43 | 0.48 | 0.66 | 0.73 | sur | 0.40 | 0.49 | 0.68 | 0.74 | sus | 0.39 | 0.51 | 0.70 | 0.77 |
| swe | 0.45 | 0.55 | 0.72 | 0.78 | swg | 0.48 | 0.57 | 0.73 | 0.79 | swh | 0.47 | 0.55 | 0.72 | 0.78 |
| swk | 0.45 | 0.57 | 0.75 | 0.80 | swp | 0.44 | 0.51 | 0.69 | 0.75 | sxb | 0.44 | 0.51 | 0.69 | 0.76 |
| sxn | 0.45 | 0.50 | 0.68 | 0.75 | syc | 0.49 | 0.53 | 0.70 | 0.76 | szb | 0.40 | 0.34 | 0.52 | 0.60 |
| tab | 0.45 | 0.51 | 0.69 | 0.75 | tac | 0.42 | 0.28 | 0.45 | 0.53 | taj | 0.47 | 0.46 | 0.65 | 0.72 |
| tam | 0.43 | 0.60 | 0.77 | 0.82 | taq | 0.44 | 0.48 | 0.66 | 0.73 | tar | 0.42 | 0.38 | 0.58 | 0.65 |
| tat | 0.46 | 0.56 | 0.74 | 0.79 | tav | 0.47 | 0.32 | 0.51 | 0.60 | taw | 0.39 | 0.33 | 0.52 | 0.60 |
| tbc | 0.47 | 0.38 | 0.58 | 0.65 | tbg | 0.41 | 0.35 | 0.54 | 0.62 | tbl | 0.43 | 0.45 | 0.64 | 0.72 |
| tbo | 0.45 | 0.46 | 0.66 | 0.73 | tby | 0.44 | 0.51 | 0.69 | 0.75 | tbz | 0.44 | 0.40 | 0.60 | 0.67 |
| tca | 0.44 | 0.41 | 0.60 | 0.67 | tcc | 0.49 | 0.52 | 0.69 | 0.76 | tcs | 0.40 | 0.43 | 0.63 | 0.70 |
| tcz | 0.45 | 0.52 | 0.69 | 0.76 | tdt | 0.45 | 0.53 | 0.71 | 0.77 | tee | 0.46 | 0.42 | 0.60 | 0.67 |
| tel | 0.46 | 0.54 | 0.72 | 0.78 | tem | 0.46 | 0.56 | 0.73 | 0.79 | teo | 0.49 | 0.57 | 0.74 | 0.79 |
| ter | 0.45 | 0.43 | 0.63 | 0.70 | tfr | 0.40 | 0.40 | 0.59 | 0.67 | tgk | 0.46 | 0.59 | 0.75 | 0.80 |
| tgl | 0.46 | 0.55 | 0.72 | 0.78 | tgp | 0.47 | 0.44 | 0.64 | 0.71 | tha | 0.49 | 0.54 | 0.72 | 0.78 |
| thk | 0.46 | 0.52 | 0.70 | 0.76 | tih | 0.47 | 0.51 | 0.69 | 0.76 | tik | 0.45 | 0.41 | 0.60 | 0.67 |
| tim | 0.44 | 0.30 | 0.49 | 0.57 | tir | 0.46 | 0.59 | 0.76 | 0.81 | tku | 0.42 | 0.44 | 0.63 | 0.70 |
| tlb | 0.47 | 0.51 | 0.69 | 0.75 | tlf | 0.44 | 0.30 | 0.48 | 0.56 | tlh | 0.50 | 0.58 | 0.75 | 0.80 |
| tna | 0.45 | 0.40 | 0.59 | 0.67 | tnn | 0.42 | 0.38 | 0.58 | 0.66 | tob | 0.43 | 0.39 | 0.59 | 0.67 |
| toc | 0.41 | 0.40 | 0.58 | 0.66 | toh | 0.49 | 0.54 | 0.71 | 0.77 | toj | 0.41 | 0.35 | 0.53 | 0.61 |
| too | 0.42 | 0.46 | 0.65 | 0.72 | top | 0.41 | 0.36 | 0.53 | 0.61 | tos | 0.40 | 0.39 | 0.58 | 0.65 |
| tpi | 0.42 | 0.48 | 0.68 | 0.75 | tpm | 0.41 | 0.31 | 0.51 | 0.59 | tpp | 0.45 | 0.50 | 0.68 | 0.74 |
| tpt | 0.44 | 0.45 | 0.63 | 0.70 | tpz | 0.46 | 0.38 | 0.58 | 0.66 | tqb | 0.39 | 0.29 | 0.47 | 0.55 |
| trc | 0.40 | 0.32 | 0.51 | 0.60 | trn | 0.43 | 0.47 | 0.66 | 0.73 | trq | 0.44 | 0.38 | 0.57 | 0.65 |
| tsn | 0.46 | 0.57 | 0.74 | 0.80 | tsz | 0.44 | 0.44 | 0.63 | 0.70 | ttc | 0.43 | 0.44 | 0.63 | 0.70 |
| tte | 0.45 | 0.46 | 0.65 | 0.72 | tuc | 0.42 | 0.39 | 0.58 | 0.66 | tue | 0.44 | 0.44 | 0.62 | 0.70 |
| tuf | 0.40 | 0.34 | 0.52 | 0.60 | tui | 0.43 | 0.38 | 0.57 | 0.65 | tuk | 0.46 | 0.52 | 0.70 | 0.76 |
| tum | 0.47 | 0.59 | 0.76 | 0.81 | tuo | 0.45 | 0.39 | 0.58 | 0.66 | tur | 0.45 | 0.55 | 0.72 | 0.78 |
| twi | 0.48 | 0.45 | 0.63 | 0.70 | twu | 0.44 | 0.47 | 0.66 | 0.73 | txu | 0.43 | 0.31 | 0.50 | 0.58 |
| tyv | 0.43 | 0.54 | 0.72 | 0.78 | tzh | 0.45 | 0.52 | 0.70 | 0.77 | tzj | 0.44 | 0.44 | 0.63 | 0.70 |
| tzo | 0.46 | 0.51 | 0.68 | 0.75 | ubr | 0.48 | 0.52 | 0.70 | 0.76 | ubu | 0.44 | 0.39 | 0.57 | 0.64 |
| udu | 0.46 | 0.44 | 0.62 | 0.70 | uig | 0.46 | 0.52 | 0.70 | 0.76 | ukr | 0.48 | 0.57 | 0.74 | 0.79 |
| upv | 0.44 | 0.42 | 0.61 | 0.68 | ura | 0.43 | 0.35 | 0.53 | 0.61 | urb | 0.39 | 0.33 | 0.53 | 0.61 |
| urd | 0.47 | 0.57 | 0.74 | 0.80 | urk | 0.45 | 0.49 | 0.69 | 0.76 | usa | 0.43 | 0.40 | 0.59 | 0.66 |
| usp | 0.38 | 0.46 | 0.65 | 0.72 | uvl | 0.42 | 0.39 | 0.59 | 0.67 | uzb | 0.44 | 0.57 | 0.74 | 0.80 |
| vag | 0.42 | 0.47 | 0.66 | 0.73 | ven | 0.47 | 0.55 | 0.72 | 0.78 | vie | 0.49 | 0.50 | 0.68 | 0.75 |
| viv | 0.44 | 0.47 | 0.66 | 0.72 | vmy | 0.44 | 0.49 | 0.68 | 0.74 | vun | 0.46 | 0.54 | 0.71 | 0.77 |
| vut | 0.45 | 0.47 | 0.66 | 0.73 | waj | 0.43 | 0.41 | 0.60 | 0.67 | wal | 0.48 | 0.53 | 0.71 | 0.77 |
| wap | 0.44 | 0.44 | 0.63 | 0.70 | war | 0.48 | 0.60 | 0.77 | 0.82 | way | 0.42 | 0.43 | 0.62 | 0.69 |
| wbm | 0.47 | 0.45 | 0.64 | 0.71 | wbp | 0.45 | 0.29 | 0.47 | 0.55 | wca | 0.41 | 0.33 | 0.52 | 0.60 |
| wer | 0.43 | 0.38 | 0.57 | 0.65 | whk | 0.46 | 0.49 | 0.67 | 0.74 | wiu | 0.44 | 0.36 | 0.55 | 0.63 |
| wmw | 0.49 | 0.55 | 0.72 | 0.78 | wnc | 0.40 | 0.34 | 0.53 | 0.60 | wnu | 0.41 | 0.29 | 0.48 | 0.57 |
| wob | 0.43 | 0.37 | 0.56 | 0.64 | wol | 0.45 | 0.52 | 0.69 | 0.76 | wos | 0.40 | 0.34 | 0.53 | 0.60 |
| wrs | 0.43 | 0.39 | 0.58 | 0.65 | wsk | 0.43 | 0.35 | 0.54 | 0.62 | wuv | 0.43 | 0.51 | 0.70 | 0.76 |
| wwa | 0.44 | 0.46 | 0.65 | 0.72 | xal | 0.46 | 0.49 | 0.67 | 0.74 | xav | 0.45 | 0.33 | 0.53 | 0.61 |
| xbr | 0.45 | 0.54 | 0.72 | 0.78 | xed | 0.43 | 0.49 | 0.68 | 0.75 | xho | 0.47 | 0.55 | 0.72 | 0.78 |
| xla | 0.42 | 0.37 | 0.56 | 0.64 | xon | 0.44 | 0.49 | 0.68 | 0.74 | xrb | 0.42 | 0.36 | 0.56 | 0.64 |
| xsi | 0.43 | 0.37 | 0.56 | 0.65 | xsm | 0.44 | 0.43 | 0.63 | 0.71 | xsu | 0.43 | 0.33 | 0.52 | 0.60 |
| xtd | 0.43 | 0.35 | 0.54 | 0.62 | xtm | 0.40 | 0.39 | 0.59 | 0.66 | xuo | 0.42 | 0.41 | 0.60 | 0.68 |
| yaa | 0.44 | 0.33 | 0.51 | 0.58 | yad | 0.45 | 0.42 | 0.60 | 0.67 | yal | 0.47 | 0.55 | 0.72 | 0.78 |
| yam | 0.42 | 0.39 | 0.59 | 0.67 | yan | 0.45 | 0.39 | 0.59 | 0.66 | yaq | 0.44 | 0.40 | 0.59 | 0.66 |
| yby | 0.43 | 0.34 | 0.53 | 0.61 | ycn | 0.44 | 0.32 | 0.51 | 0.59 | yle | 0.40 | 0.25 | 0.43 | 0.51 |
| yli | 0.42 | 0.35 | 0.54 | 0.62 | yml | 0.42 | 0.41 | 0.60 | 0.67 | yon | 0.40 | 0.44 | 0.63 | 0.70 |
| yor | 0.45 | 0.50 | 0.67 | 0.74 | yrb | 0.44 | 0.31 | 0.50 | 0.58 | yre | 0.45 | 0.36 | 0.56 | 0.64 |
| yss | 0.41 | 0.33 | 0.52 | 0.60 | yua | 0.46 | 0.54 | 0.72 | 0.78 | yuj | 0.38 | 0.39 | 0.57 | 0.65 |
| yut | 0.43 | 0.43 | 0.63 | 0.70 | yuw | 0.43 | 0.41 | 0.61 | 0.68 | yuz | 0.40 | 0.40 | 0.58 | 0.65 |
| yva | 0.47 | 0.45 | 0.65 | 0.72 | zaa | 0.41 | 0.40 | 0.61 | 0.68 | zab | 0.44 | 0.46 | 0.64 | 0.71 |
| zac | 0.40 | 0.42 | 0.62 | 0.69 | zad | 0.40 | 0.45 | 0.64 | 0.71 | zae | 0.45 | 0.45 | 0.64 | 0.71 |
| zai | 0.45 | 0.42 | 0.62 | 0.69 | zam | 0.43 | 0.35 | 0.55 | 0.63 | zao | 0.43 | 0.44 | 0.63 | 0.70 |
| zar | 0.43 | 0.47 | 0.66 | 0.73 | zas | 0.40 | 0.44 | 0.62 | 0.69 | zat | 0.44 | 0.46 | 0.65 | 0.72 |
| zav | 0.41 | 0.38 | 0.57 | 0.65 | zaw | 0.43 | 0.46 | 0.65 | 0.72 | zca | 0.44 | 0.36 | 0.55 | 0.63 |
| zho | 0.49 | 0.60 | 0.77 | 0.82 | zia | 0.43 | 0.35 | 0.54 | 0.62 | ziw | 0.50 | 0.56 | 0.73 | 0.79 |
| zom | 0.48 | 0.48 | 0.66 | 0.73 | zos | 0.44 | 0.44 | 0.62 | 0.69 | zpc | 0.41 | 0.39 | 0.58 | 0.65 |
| zpi | 0.45 | 0.47 | 0.66 | 0.73 | zpl | 0.45 | 0.41 | 0.60 | 0.68 | zpm | 0.42 | 0.34 | 0.53 | 0.60 |
| zpo | 0.46 | 0.41 | 0.60 | 0.68 | zpq | 0.37 | 0.39 | 0.58 | 0.66 | zpt | 0.43 | 0.43 | 0.63 | 0.71 |
| zpu | 0.43 | 0.38 | 0.57 | 0.65 | zpv | 0.43 | 0.40 | 0.59 | 0.67 | zpz | 0.46 | 0.38 | 0.57 | 0.65 |
| zsm | 0.47 | 0.56 | 0.73 | 0.79 | zsr | 0.46 | 0.48 | 0.67 | 0.73 | ztq | 0.39 | 0.43 | 0.63 | 0.71 |
| zty | 0.45 | 0.45 | 0.64 | 0.71 | zul | 0.46 | 0.55 | 0.72 | 0.78 | zyp | 0.45 | 0.49 | 0.68 | 0.74 |

Table 14: Transfer performance using other languages as the train/query language (Part V).