# OpenReview forum: "Crosslingual Transfer Learning for Low-Resource Languages Based on Multilingual Colexification Graphs"
_EMNLP/2023/Conference — EMNLP 2023 Findings_

### Official Review · Reviewer_A1sh · 2023-08-04

**Soundness:** 4

**Excitement:**

3: Ambivalent: It has merits (e.g., it reports state-of-the-art results, the idea is nice), but there are key weaknesses (e.g., it describes incremental work), and it can significantly benefit from another round of revision. However, I won't object to accepting it if my co-reviewers champion it.

**Paper Topic And Main Contributions:**

This paper attempts to build multilingual word embeddings for a large number of languages (1335 to be precise) by exploiting colexification patterns i.e. the phenomenon where a language expresses two different meaning with the same lexical form. The central contribution of this paper is that it builds a large-scale undirected colexification graph where each node is a concept (i.e. a lemma in English, for the purposes of this work), and the edge between two nodes indicates how many languages co-lexify the two concepts. Such a graph is automatically constructed from a large multi-way parallel corpus using a two step alignment process on the corpus (using the process from Liu et al., ACL 2023). After construction, this graph is modified to represent colexification between concepts and n-grams, by replacing each edge in the graph with a new set of edges that capture the ngrams via which the concepts are co-lexified.

This graph is then used to train a multilingual embedding model using node2vec, with the hypothesis that colexification is a critical signal for cross-lingual transfer. Evaluation is then carried out first in an intrinsic way, by identifying whether automatically constructed graph captures known set of colexification patterns. Extrinsic evaluation focuses on comparing different embedding models (trained on the same corpus) on 3 different tasks---round trip translation, passage retrieval, document similarity---and the proposed embeddings are shown to be more accurate across the board.

There is additional analysis to identify the nature of the graph created as well an the ability to use a language other than English to seed the cross-lingual transfer process.

**Questions For The Authors:**

A. Can you comment on how your work is similar to (and also different from) the pivoting approach used by e.g. [1]? It seems to me that there is some overlap in the idea of colexification and the idea that paraphrases in a single language can be identfied by pivoting in a different language.
B. In the very interesting analysis that you have in Appendix B, did you observe more semantically organized errors in colexification? For instance, [2] shows that automatically identified bilingual lexicons have a variety of relations exhibited in them. I wonder if colexification graphs also have such semantic errors.



[1] - https://aclanthology.org/P05-1074.pdf
[2] - https://dl.acm.org/doi/10.1145/2050100.2050102


**Reasons To Accept:**

* Through experimentation: The paper does a very controlled and careful set of experiments to support its claim. For instance, all baselines are trained on the same corpus, with the same dimensionality, etc. to ensure there is no arbitrary variance.
* Colexicifcation for language transfer is an interesting strategy that has not been explored by prior work on multilingual embeddings. This work does a very through investigation on this phenomenon, proposes a technique to automatically build a graph that identifies colexfication for > 1000 languages, and effectively uses it for cross-lingual transfer. That, in my opinion, is a series of contributions that are unique to this field.

**Reasons To Reject:**

No strong reason to reject. I believe the paper could have been enhanced had it been compared with non-control baselines also (e.g. models that have access to more data) to identify how far away this approach is from more powerful lexical representations, trained on more data. However, that is not required and the paper tells a complete and coherent story with its experiments.

**Reproducibility:**

4: Could mostly reproduce the results, but there may be some variation because of sample variance or minor variations in their interpretation of the protocol or method.

**Reviewer Confidence:**

4: Quite sure. I tried to check the important points carefully. It's unlikely, though conceivable, that I missed something that should affect my ratings.

---

> ### Author Rebuttal · Authors · 2023-08-27
>
> We appreciate your time and effort in reviewing our paper. We would like to address your concern as follows:
>
> - ***"I believe the paper could have been enhanced ... with its experiments."***
>
> Thanks for mentioning this. The major problem is that most models only support a small portion of the languages that our model supports. As we wanted to compare on as many languages as possible, we did not include other non-control baselines in the current version. We are considering including the non-control baseline Glot500 [1] (a continue-trained model of XLM-R on more than 500 languages) for comparison in the camera-ready version. According to their results, for extremely low-resourced languages (very likely unseen by the transformer-based model), XLM-R or Glot500 perform far behind multilingual embeddings.
>
>
> - ***"A. Can you comment on how your work is similar to ... a different language."***
>
> Both our method and [2] have a two-step working flow:
>
> - [2] looks at what foreign language phrases the English translates to, and finds all occurrences of those foreign phrases (this roughly corresponds to the forward pass in our method to find target-language ngrams)
> - [2] then looks back at what other English phrases they (those foreign phrases) translate to (this roughly corresponds to the backward pass to find English concepts).
>
> The major differences are that: [2] works on a token-level instead of ngrams; [2] needs a token-level word/phrase aligner to get the initial alignments, while Conceptualizer (used in our method) itself integrates a $\chi^2$-based concept-ngram alignment method. Actually, this kind of idea is also similar to the semantic mirror [3], a method to explore semantic relations using translational data. Thanks for mentioning the paper and we will cite the paper in our camera-ready version.
>
> - ***"B. In the very interesting analysis that you have in Appendix B ... such semantic errors."***
>
> This is a very good question. In our setting, the extraction of colexification patterns is formed similarly to bilingual lexicon identification, where we first identify target-language ngrams that correspond to an English concept, and then English concepts that correspond to those identified ngrams. Therefore, our method is not immune to semantic errors, such as antonyms, hypernyms, or hyponyms. In our manual inspection, we found **free-translation** and **co-occurrence** are the two major reasons for many types of semantic errors, as the examples shown in Appendix B. This finding matches the statistics in [4], which shows that "Related" makes up the major part of the semantic errors. Thanks for mentioning the paper and will cite this in our camera-ready version.
>
>
> [1] https://aclanthology.org/2023.acl-long.61.pdf
>
> [2] https://aclanthology.org/P05-1074.pdf
>
> [3] https://clara.w.uib.no/files/2011/06/semmirrors.pdf
>
> [4] https://dl.acm.org/doi/10.1145/2050100.2050102

---

### Official Review · Reviewer_knrQ · 2023-08-11

**Soundness:** 4

**Excitement:**

4: Strong: This paper deepens the understanding of some phenomenon or lowers the barriers to an existing research direction.

**Paper Topic And Main Contributions:**

UPDATE: I thank the authors for their clarifications.

This paper proposes to build multilingual graphs (ColexNet and ColexNet+) from colexification patterns that have been automatically identified on a corpus of 1,334 languages (the Bible). These graphs are shown to achieve high recall at capturing crosslingual colexifications. The authors propose to obtain multilingual embeddings from one of these graphs and evaluate them on roundtrip translation, verse retrieval and verse classification. Results show that the proposed embeddings perform better than a number of existing alternatives. In one experiment, transfer learning is performed from every language to every language.

I do not have any big criticism for this paper, I think it should be accepted.

Perhaps it would be nice if some precision-like measure could be calculated for the colexification identification task, but I understand this would not be straightforward.

**Questions For The Authors:**

- A. The working of the Conceptualizer reminds me of Bannard and Callison-Burch's pivot method to discover paraphrases (https://aclanthology.org/P05-1074.pdf). I guess that the lambda threshold limits, too, the presence of paraphrases, but I would have expected to see some comments on this. Could you briefly compare the methods or add some words on this?

**Reasons To Accept:**

It is very well-written, the topic is relevant, it addresses a very large number of languages, the proposed approach is simple and sound and the experiments are interesting. Additionally, there are interesting and extensive analyses in the Appendices. The limitations are appropriately addressed and acknowledged in the corresponding section.

**Reasons To Reject:**

I don't see any big weaknesses.

**Reproducibility:**

4: Could mostly reproduce the results, but there may be some variation because of sample variance or minor variations in their interpretation of the protocol or method.

**Reviewer Confidence:**

3: Pretty sure, but there's a chance I missed something. Although I have a good feel for this area in general, I did not carefully check the paper's details, e.g., the math, experimental design, or novelty.

**Typos Grammar Style And Presentation Improvements:**

- l240 "l = 3 iterations". -> "l" is already used for language. From the Appendix I'd assume it should be "M".

- I found the CLIQUE embeddings description a bit confusing.

- Could you make Figure 2's font larger?

---

> ### Author Rebuttal · Authors · 2023-08-27
>
> Thank you very much for your review and positive comments. We would like to answer your question as follows:
>
> - ***"Perhaps it would be nice if some precision-like measure ... not be straightforward."***
>
> Yes, directly using precision might not be the best option as it can underestimate the performance. This is because the colexification patterns in CLICS itself are not comprehensive and many patterns included in ColexNet can actually be correct (please refer to Appendix B). Therefore we use #aw_colex, which has a similar function to the precision. This metric can better and more directly reflect how the exact number of "wrong" patterns per concept changes with respect to \lambda.
>
> - ***"A. The working of the Conceptualizer reminds me of Bannard and Callison-Burch's pivot method ... some words on this?"***
>
> [1] also has a two-step working flow as Conceptualizer: (1) look at what foreign language phrases the English translates to and find all occurrences of those foreign phrases (this roughly corresponds to the forward pass in Conceptualizer) (2) then look back at what other English phrases they translate to (this roughly corresponds to the backward pass).
>
> The major differences are that: [1] works on a token level instead of ngrams; [1] needs a token-level word/phrase aligner to get the initial alignments, while Conceptualizer itself integrates a $\chi^2$-based concept-ngram alignment method. Thanks for mentioning the paper and we will cite it in our camera-ready version.
>
> The $\lambda$ threshold in our method does not directly influence the presence of paraphrases, as we have a pre-defined concept pool (a set of English lemmata) as the candidates on the English side for the two-step alignment. We agree that there might be some paraphrases e.g., <prison> and <jail>, in the concept pool. We simply regard them as different concepts under such cases. The lambda threshold mainly influences the quality of the identified colexification patterns. As we discussed in our paper, if the threshold \lambda is set larger, fewer concepts and fewer edges will be included in ColexNet. If two concepts are colexified in many languages, we can be more certain of the colexification relation between the two concepts.
>
> We will fix the typos and further improve the presentation of the paper.
>
> [1] https://aclanthology.org/P05-1074.pdf

---

### Official Review · Reviewer_7BGY · 2023-08-12

**Soundness:** 4

**Excitement:**

3: Ambivalent: It has merits (e.g., it reports state-of-the-art results, the idea is nice), but there are key weaknesses (e.g., it describes incremental work), and it can significantly benefit from another round of revision. However, I won't object to accepting it if my co-reviewers champion it.

**Paper Topic And Main Contributions:**

This paper proposes an approach to build a multilingual colexification graph from the Parallel Bible Corpus. The resulting English-concept-centered colexification graph ColexNet is extended into the multilingual ColexNet+, which contains concept-realizing ngrams as intermediate nodes between abstract concepts. ColexNet is shown to robustly identify common colexification patterns to the gold standard CLICS. ColexNet+ is used to train multilingual embeddings which outperform multilingual baselines on 3 diverse downstream tasks of roundtrip translation, verse retrieval and verse classification.

**Reasons To Accept:**

ColexNet and ColexNet+ provide valuable assets to the study of multilingual colexification and linguistic typology, and the proposed data-driven algorithm is shown to be well-aligned with the manually constructed gold standard CLICS for English-stemming concepts. The algorithm proposed is replicable and can be extended to any parallel corpus. The embeddings produced from ColexNet+ potentially hold value for downstream NLP tasks in very low-resourced languages. The table of experimental results between every pair of languages in the appendix is valuable for ranking the potential of the approach for many individual extremely low-resourced languages.

**Reasons To Reject:**

The experiments section, while thorough, has the following weaknesses:

i) The tasks are either not very representative of important downstream NLP applications of word embeddings, or are only evaluated within the context of PBC. For example, the success of the embeddings on roundtrip word translation does not provide insight into to their success on a task like low-resource machine translation. The verse classification task, while based on a recent multilingual text classification benchmark, uses a rather arbitrary set of classes which seem neither comprehensive nor mutually exclusive topics for Bible verses.

ii) The baselines for multilingual embeddings do not include Transformer-based models. This is understandable for extremely low-resourced languages, but it would be useful to see a comparison of ColexNet+ embeddings with multilingual transformer encoders such as XLM-R (Conneau et al. 2020). A task such as verse retrieval seems like a good place to do this, at least for the subset of language pairs that are represented in the 88 languages on which XLM-R was trained. Can the ColexNet+ embeddings be shown to outperform multilingual Transformer encoders for extremely low-resourced languages? An answer to this question is consequential to the applicability of the proposed multilingual embeddings on downstream tasks.

**Reproducibility:**

5: Could easily reproduce the results.

**Reviewer Confidence:**

3: Pretty sure, but there's a chance I missed something. Although I have a good feel for this area in general, I did not carefully check the paper's details, e.g., the math, experimental design, or novelty.

---

> ### Author Rebuttal · Authors · 2023-08-27
>
> Thank you very much for your review and thoughtful feedback. We would like to address your concerns as follows:
>
> - ***"i) The tasks are either not very representative of important downstream ... neither comprehensive nor mutually exclusive topics for Bible verses."***
>
> As we acknowledged in the Limitation section, there are few evaluation datasets that cover such a wide range of languages including low-resource languages in the community. Therefore we have to resort to some in-domain tasks to evaluate the quality of the embeddings and compare with other embedding counterparts. The evaluation shows that our embeddings outperform all other embeddings on the same tasks, which can verify the effectiveness of the introduction of colexification into NLP. Indeed, good performance on word-level translation tasks does not guarantee good machine translation performance, since machine translation additionally requires the ability of language generation. Nevertheless, embeddings are an essential part of modern NLP applications and there are already some works showing how well-aligned multilingual embeddings, combined with Transformer-based language models, can boost the performance on various downstream tasks, including low-resource machine translation [1]. Therefore we would expect our work can similarly provide some insights for future works involving low-resource languages in the era of LLMs. Although the verse classification benchmark (the set of classes as well as the verses belonging to each class are carefully selected according to [2]) is also limited to the Bible, it is the only dataset that supports most of the languages that we support, making it possible to do a fair comparison between multiple methods on a large-scale set of languages.
>
> - ***"ii) The baselines for multilingual embeddings do not include Transformer-based ... applicability of the proposed multilingual embeddings on downstream tasks."***
>
> This is a very good point. Except for the BOW baseline, all other baselines are embeddings learned from the same data and therefore they are fully comparable. Transformer-based models like XLM-R are pre-trained on unannotated corpus and do not explicitly learn the lexical-level alignment knowledge. As a result, the transfer performance, especially for the low-resource languages, can be very bad in these tasks. A recent paper [3] confirms this. They reported XLM-R (base and large) and Glot500 (a continue-trained model of XLM-R on more than 500 languages) on several tasks including verse retrieval. For many extremely low-resourced languages, the results are far less than our multilingual embeddings. We will include this comparison in the camera-ready version. As an insight, training an LLM considerably consumes resources and it only works if enough data is available. On the other hand, training embeddings is much more efficient and we can get good performance if we use small-size parallel data to explicitly extract some lexical-level knowledge (like what we did). Furthermore, it is worth exploring if such lexical knowledge encoded in the multilingual embeddings could be used to improve the crosslinguality of an LLM, therefore boosting its transfer learning.
>
>
> [1] https://aclanthology.org/2021.naacl-main.16.pdf
>
> [2] https://arxiv.org/pdf/2305.08487.pdf
>
> [3] https://aclanthology.org/2023.acl-long.61.pdf

---

### Official Review · Reviewer_YCED · 2023-08-16

**Soundness:** 4

**Excitement:**

3: Ambivalent: It has merits (e.g., it reports state-of-the-art results, the idea is nice), but there are key weaknesses (e.g., it describes incremental work), and it can significantly benefit from another round of revision. However, I won't object to accepting it if my co-reviewers champion it.

**Paper Topic And Main Contributions:**

This paper deals with the automatic construction of a colexification graph and its application to the training of multilingual word representations. The authors obtained the following results:
1) Using Conceptualizer [1] and the Parallel Bible Corpus, the authors automatically constructed a colexification graph for 1,335 languages.
2) The resulting graph was evaluated on the CLICS dataset.
3) The authors propose a modified version of a colexification graph with nodes of target language n-grams between nodes of English concepts.
4) Using a modified version of the colexification graph, the authors trained aligned multilingual word representations.
5) The resulting multilingual representations were evaluated in three different tasks in a cross-lingual scenario.

[1] [A Crosslingual Investigation of Conceptualization in 1335 Languages](https://aclanthology.org/2023.acl-long.726) (Liu et al., ACL 2023)

**Questions For The Authors:**

A) Is there any other work on the automatic construction of colexification graphs? Is it possible to use them to train word embeddings?
B) Since your work is about multilingual word embeddings, is it possible to learn word embeddings based on the gold colexification patterns (e.g. CLICS)? Is there a reference or comparison to your results?
C) Why have you decided to compute only the Recall metric on CLICS and to propose the new metric #aw_colex instead of Precision?
D) What are the advantages and disadvantages of using lemmas for English concepts while using n-grams for other languages?
E) Is there a correlation between performance on CLICS in low/high resource languages and the quality of transfer learning?
F) The paper on Conceptualizer [1] also proposes a new metric of similarity between languages based on colexification patterns. Is there a correlation between this metric and the quality of transfer learning?

**Reasons To Accept:**

1) This paper extends previous results [1], where Conceptualizer was applied and evaluated on a small set of manually annotated concepts and languages.
2) The authors propose a new method for training multilingual word representations that are competitive in downstream tasks.
3) This work could improve our understanding of the mechanism of cross-lingual transfer learning.

**Reasons To Reject:**

My main concerns are about the focus of the paper and the presentation of the results (see section "Typos, Grammar Style and Presentation Improvements").
1) The title of the paper is about "cross-lingual transfer learning", but big part of the paper is about graph construction. Basically it's a conversion of Conceptualizer output into a new format. I think the paper should focus more on multilingual embeddings and transfer learning or be renamed to something like "evaluation of Conceptualizer on the large number of concepts".
2) The authors propose two methods for constructing a colexification graph: ColexNet and ColexNet+. It seems that the only purpose of the first method (ColexNet) is the evaluation on CLICS, which can also be done on ColexNet+. I suppose that the graph construction part can be reduced to one method.

**Reproducibility:**

4: Could mostly reproduce the results, but there may be some variation because of sample variance or minor variations in their interpretation of the protocol or method.

**Reviewer Confidence:**

4: Quite sure. I tried to check the important points carefully. It's unlikely, though conceivable, that I missed something that should affect my ratings.

**Typos Grammar Style And Presentation Improvements:**

During reviewing I had to read a paper about Conceptualizer [1] because this paper uses some concepts from the previous work and poorly describes the main tool (Conceptualizer). E.g. it was not clear precisely what are colexification patterns, focal concepts, ngrams, dollar sign, input and output of the Conceptualizer. Ideally it would be nice to see an image how graph from Conceptualizer converts to the ColexNet+ graph.

There are some inaccuracies in the description of the algorithm:
1) Conceptualizer returns not only n-gram but also a verse id and language id, which are not mentioned in the algorithm or its description
2) Initially there are no edges in the graph. So what is “e” for which you initialize edges on the line 4 of the algorithm?
3) Somewhere you write “w_c(e)” with one pair brackets, somewhere with two pairs ““w_c((e))”.
4) Line 13 — missed closing bracket “wn((f, c)”

Line 240 of the paper — use “l” as the maximum number of the iterations. I suppose it should be “M”.

What is for “d_tx = 1” in the line 333 if it is not possible?

In Section 5.2 It would be nice to describe the results of transfer learning not only from English but also from the other languages.

---

> ### Author Rebuttal · Authors · 2023-08-27
>
> Thank you very much for your thorough review and valuable comments. We would like to address your concerns and questions as follows:
>
> - ***"The title of the paper is about "cross-lingual transfer learning" ... number of concepts."***
>
> Our focus in this paper is large-scale **crosslingual transfer learning**, especially for low-resource languages. We achieve this by leveraging multilingual colexification graphs, which are constructed from the colexification patterns extracted from a parallel corpus. Thus, the multilingual graph is a very important part of our paper and we tried to be as detailed as possible when introducing the graph construction. Of course, the crosslingual transfer-learning evaluation is vital as well. Therefore we also conducted extensive experiments including two evaluations (verse retrieval and classification), which really tested the zero-shot crosslingual transferability. In addition to the evaluation in the main content, we also include a detailed analysis in Appendix E and F (English and beyond English-centric transfer), which shows the performance by using **any** language as train/query language. We will move some analysis of the transfer learning to the main content in the camera-ready version.
>
> - ***"The authors propose two methods for constructing a colexification ... to one method."***
>
> Though, as you said, the evaluation on CLICS can also be done on ColexNet+ (then we have to filter out the ngram nodes), ColexNet serves as a graph where we can **easily and efficiently** analyze its structure because it is lightweight (5870 nodes at most) and homogeneous in nodes (just concept nodes). Furthermore, we explored the centralities and communities in ColexNet and obtained some interesting findings (please refer to Section 5.1, Appendix C and D), which cannot be done directly on ColexNet+. In addition, the aims of the two graphs are different: ColexNet is used to explore the colexification patterns across languages; ColexNet+ is used to train multilingual embeddings for crosslingual transfer.
>
> - ***"A) Is there any other work on ... train word embeddings?"***
>
> There are a few works on constructing colexification graphs, such as [1] and [2] (mentioned in Introduction and Related Works). However, they are not "automatic". These methods rely on colexification patterns from external hand-curated lexica, while our method extracts the patterns directly from a parallel corpus. These two works also train embeddings, but they focus on lexical-level tasks such as semantic word similarity and word sense disambiguation, while we focus on crosslingual transfer learning in NLP. Their embeddings cannot easily be used in tasks such as sentence retrieval or classification, because the morphological changes are very frequent in many languages, while the tokens included in these embeddings are mostly in dictionary form.
>
> - ***"B) Since your work is about multilingual word embeddings ... to your results?"***
>
> This is a good idea and we did some preliminary experiments earlier. The major problem is the coverage: for many low-resource languages that we support, CLICS only supports less than a hundred concepts, which will simply lead to very bad performance. But it would be a good future direction to combine the gold colexification patterns and automatically extracted patterns.
>
> - ***"C) Why have you decided to compute only the Recall ... instead of Precision?"***
>
> The recall metric is more interesting to us because we want to know how many ground-truth colexification patterns in CLICS are included in ColexNet, e.g., if there is an edge between "hand" and "arm" in CLICS, we want an edge between these two concepts presenting in ColexNet as well. Precision might not be a good metric because patterns in CLICS are not comprehensive and many patterns included in ColexNet can actually be correct (please refer to Appendix B). Directly using precision then underestimates the performance. Therefore we use #aw_colex, which has a similar function to the precision, but can better and more directly reflect how the exact number of "wrong" patterns per concept changes with respect to $\lambda$.
>
> - ***"D) What are the advantages and disadvantages ... for other languages?"***
>
> The major advantages are that: (1) lemmatizer tools in English perform pretty well so we could use them to eliminate redundancy in concept representations, e.g., we will have "belly" instead of both "belly" and "bellies" to represent <belly> concept. (2) there are no available tokenization/lemmatization tools for most low-resource languages, therefore using ngrams gives **more freedom** when selecting best-matching character combinations. One disadvantage is that we observe some identified ngrams can sometimes include some redundant characters, e.g., "鸽子的" (pigeon in Chinese) instead of "鸽子" (pigeon), mainly for languages without separators between words.
>
> - ***"E) Is there a correlation between performance ... quality of transfer learning?"***
>
> In Appendix F, we closely looked at how low/high resource languages when serving as train/query language influence the transfer performance. Here are some takeaways: (1) high-resource languages often mean higher quality of the extracted colexification patterns. And the quality is positively correlated with the transfer performance; (2) the model can have difficulties in inducing reliable colexification patterns for some low-resource languages due to various language-specific properties, and therefore sometimes can achieve bad performance.
>
> - ***"F) The paper on Conceptualizer also proposes ... of transfer learning?"***
>
> This is a very insightful question. We guess there will be a positive correlation between the conceptual similarity and the transfer performance. We will do an additional test on this and include it in the camera-ready version.
>
> - ***"What is for “d_tx = 1” in line 333 if it is not possible?***
>
> The answer is: $d_{tx}$ can only be 0 or 2 in ColexNet+. The equation shown in the paper is the sampling strategy of Node2Vec. We explain how it is applied in our sampling in lines 335-344.
>
>
> We will fix the typos and further improve the presentation of the paper, e.g., in more detail introducing some concepts from previous works, to improve the readability.
>
>
>
> [1] https://aclanthology.org/2022.naacl-main.386/
>
> [2] https://openreview.net/pdf?id=Hn10RDGms3p

---

### Meta-Review · Area_Chair_z86T · 2023-09-24

**Recommendation:** 4

**Metareview:**

Summary (adapted from Reviewer 7BGY): This paper proposes an approach to build a multilingual colexification graph from the Parallel Bible Corpus (PBC). The resulting English-concept-centered colexification graph ColexNet is extended into the multilingual ColexNet+, which contains concept-realizing n-grams as intermediate nodes between abstract concepts. ColexNet is shown to robustly identify common colexification patterns to the gold standard CLICS. ColexNet+ is used to train multilingual embeddings which outperform multilingual baselines on 3 diverse downstream tasks of roundtrip translation, verse retrieval and verse classification.

Overall, the reviews were positive, with the “Reasons to Reject” mostly focusing on improvements in clarity in the paper and explanation of the results. Reviewer 7BGY lays out concerns about whether the colexification graph and embeddings will be of use to other tasks, and whether the Bible data is sufficient for evaluation.

These concerns are valid, but as the authors’ response makes clear regarding these comments and others, there is a tradeoff between trying to evaluate in as many languages as possible in terms of limitations of both data and models available (i.e. pretrained Transformers do not cover the number of languages needed). The authors have had to make compromises, but the paper seems sound within the scope it attempts to address.

The response period was extremely productive, and the authors provided clear responses to the reviews, all of which the reviewers acknowledged.

As came up in the discussion with Reviewer 7BGY, this paper might benefit from further discussion of its limitations with regards to the use of a Bible corpus as the primary dataset. While this is mentioned in the limitations, it receives little attention and it may be worthwhile to discuss it more explicitly and earlier in the paper.

---

### Decision · Program_Chairs · 2023-10-07

**Decision:**

Accept-Findings

**Comment:**

Summary (adapted from Reviewer 7BGY): This paper proposes an approach to build a multilingual colexification graph from the Parallel Bible Corpus (PBC). The resulting English-concept-centered colexification graph ColexNet is extended into the multilingual ColexNet+, which contains concept-realizing n-grams as intermediate nodes between abstract concepts. ColexNet is shown to robustly identify common colexification patterns to the gold standard CLICS. ColexNet+ is used to train multilingual embeddings which outperform multilingual baselines on 3 diverse downstream tasks of roundtrip translation, verse retrieval and verse classification.

Overall, the reviews were positive, with the “Reasons to Reject” mostly focusing on improvements in clarity in the paper and explanation of the results. Reviewer 7BGY lays out concerns about whether the colexification graph and embeddings will be of use to other tasks, and whether the Bible data is sufficient for evaluation.

These concerns are valid, but as the authors’ response makes clear regarding these comments and others, there is a tradeoff between trying to evaluate in as many languages as possible in terms of limitations of both data and models available (i.e. pretrained Transformers do not cover the number of languages needed). The authors have had to make compromises, but the paper seems sound within the scope it attempts to address.

The response period was extremely productive, and the authors provided clear responses to the reviews, all of which the reviewers acknowledged.

As came up in the discussion with Reviewer 7BGY, this paper might benefit from further discussion of its limitations with regards to the use of a Bible corpus as the primary dataset. While this is mentioned in the limitations, it receives little attention and it may be worthwhile to discuss it more explicitly and earlier in the paper.